# Engineered cell differentiation and sexual reproduction in probiotic and mating yeasts

Emil D. Jensen [1]✉, Marcus Deichmann [1], Xin Ma[1], Rikke U. Vilandt[1], Giovanni Schiesaro [1], Marie B. Rojek[1], Bettina Lengger [1], Line Eliasson[1], Justin M. Vento[2], Deniz Durmusoglu[2], Sandie P. Hovmand[1], Ibrahim Al'Abri [2], Jie Zhang [1], Nathan Crook [2] & Michael K. Jensen [1]✉

G protein-coupled receptors (GPCRs) enable cells to sense environmental cues and are indispensable for coordinating vital processes including quorum sensing, proliferation, and sexual reproduction. GPCRs comprise the largest class of cell surface receptors in eukaryotes, and for more than three decades the pheromone-induced mating pathway in baker's yeast *Saccharomyces cerevisiae* has served as a model for studying heterologous GPCRs (hGPCRs). Here we report transcriptome profiles following mating pathway activation in native and hGPCR-signaling yeast and use a model-guided approach to correlate gene expression to morphological changes. From this we demonstrate mating between haploid cells armed with hGPCRs and endogenous biosynthesis of their cognate ligands. Furthermore, we devise a ligand-free screening strategy for hGPCR compatibility with the yeast mating pathway and enable hGPCR-signaling in the probiotic yeast *Saccharomyces boulardii*. Combined, our findings enable new means to study mating, hGPCR-signaling, and cell-cell communication in a model eukaryote and yeast probiotics.

From the first genetic studies of yeast[1], to the discovery of heterothallism[2] and identification of genes involved in mating[3], the baker's yeast *Saccharomyces cerevisiae* (*S. cerevisiae*) has been considered a dominant model organism for studying genetics[4] and, in recent decades, heterologous G protein-coupled receptor (hGPCR) signaling[5–8]. The genome of *S. cerevisiae* encodes two GPCRs involved in mating, one for each of the two mating pheromones, namely *STE2*, which senses the α-factor produced by *MAT*α cells, and *STE3*, which senses **a**-factor produced by *MAT***a** cells[9]. *STE2* and *STE3* are not essential and can be substituted with heterologous GPCRs (hGPCRs) for health and biotechnological applications[10], including large-scale drug discovery[11], point-of-care pathogen detection[12], directed evolution of hGPCRs[13], and as sensor-actuator probiotic agents in mouse models of disease[13]. After successful hGPCR expression and G protein-coupling, cognate ligands can activate the yeast mating pathway, which constitutes an evolutionarily conserved mitogen-activated protein kinase (MAPK) pathway, recently minimized to simplify

hGPCR deorphanization and signaling studies in *S. cerevisiae*[8]. In brief, the $G_\alpha$ subunits of the heterotrimeric G protein physically bind to the native GPCRs, Ste2 and Ste3, serving as guanine-nucleotide exchange factors (GEFs). Upon ligand binding, Ste2 and Ste3 facilitate the exchange of GDP for GTP on the $G_\alpha$ subunit and subsequent $G_{\beta\gamma}$ release[10]. This, in turn, activates the MAPK cascade orchestrating major transcriptome perturbations, cell cycle arrest, polarized growth, and ultimately mating[9,14].

Despite the multitude of studies on hGPCR-coupling to the yeast mating pathway[8,10,11], there is a lack of knowledge on both systemic signaling and mating in yeast upon substitution of native pheromone GPCRs for heterologous counterparts, constraining yeast as a model organism and limiting its application within biotechnology and health. So far, omics studies of mating pathway activation have focused on native pheromone-induced explorations[14–16], while investigations of the compatibility of hGPCRs with yeast mating have been limited to substituting only one of the yeast pheromone GPCRs at the time in

[1]Novo Nordisk Foundation Center for Biosustainability, Technical University of Denmark, Kgs, Lyngby, Denmark. [2]Department of Chemical and Biomolecular Engineering, North Carolina State University, Raleigh, NC 27695, USA. ✉e-mail: emdaje@biosustain.dtu.dk; mije@biosustain.dtu.dk

*MAT***a**/*MAT*α mating pairs[5,17–19]. Such substitutions of individual yeast GPCRs with heterologous yeast pheromone receptors have permitted mating, albeit at low frequencies (≤1%)[18], while another seminal study reported that substitution of the human β2-adrenergic receptor (hβ2-AR) in place of *STE2* could not support mating[5]. Even though largely unresolved, it is expected that establishing synthetic mating in a model eukaryote would deepen our understanding of how an evolutionarily conserved MAPK signaling route, such as the yeast mating pathway, is regulated, as well as support studies of engineered cell-cell communication, mating behavior, and the development of high-throughput assays within biotechnology and medicine.

Here we compare the transcriptome of strains engineered with different hGPCRs to wild-type and sensitized yeast devoid of the negative regulator of the mating pathway, *SST2*[20], and in response to different dosages of their cognate ligands. We identify >1,000 differentially expressed genes and correlate RNA-seq data to biosensing of hGPCR ligands and mating morphology in *S. cerevisiae*, ultimately demonstrating that substituting native yeast GPCRs with signaling hGPCRs can enable efficient mating in yeast. Our study also describes a ligand-free assessment of functional hGPCR-coupling in yeast and presents a simple approach to activate dormant GPCR-signaling in the probiotic yeast *Saccharomyces boulardii*.

## Results

### Biosensing yeast strain engineering and characterization

Before studying the impact of hGPCR expression at the systems level, we first characterized biosensing in *S. cerevisiae* of adenosine (A2bR), encoded by *ADORA2B*, melatonin (MT1), encoded by *MTNR1A*, serotonin (5-HT4b), encoded by *HTR4*, and the fission yeast pheromone P-factor (Mam2), encoded by *MAM2*. The strain designs adhered to established literature[6,8,11,21], in which genes encoding hGPCRs were overexpressed from the *CCW12* promoter, while the native G$_\alpha$ subunit (*GPA1*), or a chimeric humanized G$_\alpha$ subunit (*GPA1/G$_{\alpha i2}$*), were expressed from the weak (*RNR2*), medium (*ALD6*), or strong (*PGK1*) promoters in order to differentially tune mating pathway response to hGPCR-signaling (Supplementary Fig. 1A). The strains were further deleted for the negative G protein signaling regulator *SST2* which catalyzes the exchange of GTP with GDP in the G$_\alpha$ subunit to prevent further signaling. The rationale for deleting *SST2* was to remove negative feedback regulation and increase sensitivity to the synthetic pheromones, as reported for native pheromones[22] (Supplementary Fig. 1B). The strains were additionally deleted for native pheromone receptors (*STE2* and *STE3*) and contained a substitution of *FUS1* for green fluorescent protein (GFP), which in this way became expressed from the native pheromone-inducible promoter P$_{FUS1}$ to serve as a fluorescent reporter of mating pathway signaling (Fig. 1A and Supplementary Fig. 1A). In addition, the resulting biosensing strains retained coupling to the entire mating pathway, including *FAR1* which is responsible for activation of G1 cell cycle arrest prior to mating[10]. In accordance with the previous studies[8,11,21], we observed that the P$_{FUS1}$-GFP reporter was induced in all biosensing strains across a gradient of cognate ligand dosages (Fig. 1B), except no signaling was observed for 5-HT4b in the P$_{RNR2}$-GPA1/G$_{\alpha i2}$ strain design (Fig. 1B). Importantly, and corroborating previous work on fungal GPCRs[7], hGPCRs are orthogonal across their non-cognate ligands when expressed in yeast (Supplementary Fig. 1C), while for strain designs with the strong promoter *PGK1* controlling expression of the G$_\alpha$ protein the EC50s observed for MT1 and 5-HT4b are approximately two orders of magnitude higher in yeast cells, and approximately 2-fold lower for A2bR in yeast cells, compared to mammalian cells (CHO)(Supplementary Table 1 and Supplementary Fig. 1D).

Intriguingly, in the absence of ligands, all four hGPCRs altered reporter expression up to 4-fold, solely from introduction to the P$_{RNR2}$-GPA1 background strain (Fig. 1B). We termed this phenomenon a "coupling-shift" (see *Discussion*) and observed that coupling-shifts

were also evident for six additional hGPCRs known to signal in yeast[5,7,23,24], as well as for three hGPCRs not previously shown to signal in yeast (referred here as *Zt*Ste2, *Tr*Ste2, and *Ms*Ste3) originating from pathogenic fungi, and for which cognate pheromones are currently not known (Supplementary Fig. 1E).

Thus, P$_{RNR2}$-G$_\alpha$ strain designs presented a simple pre-screen for putative coupling of hGPCRs to the yeast mating pathway, while the P$_{PGK1}$-G$_\alpha$ strain designs allowed tight control of mating pathway activation.

### Transcriptome analysis of hGPCR-signaling yeast

Activation of the pheromone-induced mating pathway in yeast evokes cell cycle arrest and cell differentiation, including expression changes in hundreds of genes[14]. To elucidate transcriptome perturbations in yeast during hGPCR-signaling, and further guide efforts to successfully coupling of hGPCRs to the yeast mating pathway, we compared transcriptomes from biosensing strains expressing A2bR, MT1, or 5-HT4b, to wild-type and sensitized (*sst2Δ*) strains with and without cognate ligand stimulation. From this analysis, we identified 1178 differentially expressed genes (DEGs) across all 20 experimental conditions (Supplementary Data 1). The wild-type and sensitized *sst2Δ* strains solely account for ~75% (878/1178) of the DEGs, while ~8% (92/1178) of the DEGs were unique to strains expressing hGPCRs (Fig. 1C). Importantly, we identified a core set of 57 DEGs that overlapped between all strains and conditions (Fig. 1C, D and Supplementary Data 1). Among this set of core DEGs, and in accordance with previous gene expression studies performed on wild-type yeast[14,25], we identified numerous mating pathway genes, including *MFA1/MFA2*, *FUS1/FUS2/FUS3*, and *FIG1*, as well as over-represented gene ontologies related to receptor signaling and conjugation, in all five strains (Fig. 1D and Supplementary Fig. 1F). Expression of individual DEGs varied greatly between individual strains, with wild-type and sensitized *sst2Δ* strains showing higher expression perturbations in response to their cognate ligand than was the case for strains expressing hGPCRs (Fig. 1D and Supplementary Data 1). For instance, neither *FUS3*, the kinase eliciting yeast mating pathway signal amplification, nor *FUS1* or *FUS2*, which are involved in membrane fusion during mating[10], were induced to the same extent in any of the biosensing yeast strains as compared to wild-type or sensitized yeast strains (Fig. 1D and Supplementary Data 1). Vice-versa, only three DEGs were shared exclusively between the three strains expressing hGPCRs. These included induction of two genes controlling expression of proteins with putative functions (*YFL065C* and *YCL121W-A*) and *STE6*, required for **a**-factor export, while 10–53 DEGs were unique to strains expressing individual hGPCRs (Fig. 1C and Supplementary Data 1). Of these, the majority of DEGs were found in samples treated with ligands, meaning that rather few DEGs were specific to expression of individual hGPCRs in yeast.

Interestingly, we noticed lower expression of chimeric *GPA1/G$_{\alpha i2}$* as compared to *GPA1* (Supplementary Data 2), although only the 5 C-terminal amino acids differed in these designs, and we identified several DEGs relevant for future engineering of biosensing and actuation, such as *AGA2, KAR4*, and *MFA1/MFA2*, which all showed >10-fold activation across all strains. Of these, *MFA1/MFA2* displayed high basal activities in biosensing strains, while *AGA2* and *KAR4* showed basal activities lower than in the wild-type strain (Supplementary Data 1).

In conclusion, our transcriptome analysis identified close to 900 DEGs shared only among the wild-type and sensitized *sst2Δ* yeasts, a core set of 57 DEGs shared between wild-type, sensitized *sst2Δ*, and all biosensing yeasts, as well as several promoters dynamically regulated during hGPCR-signaling.

### A single genetic edit enables the biosensing of probiotic yeast

Extending from our initial analyses, we next investigated if biosensing strain designs from *S. cerevisiae* could be transferred into probiotic

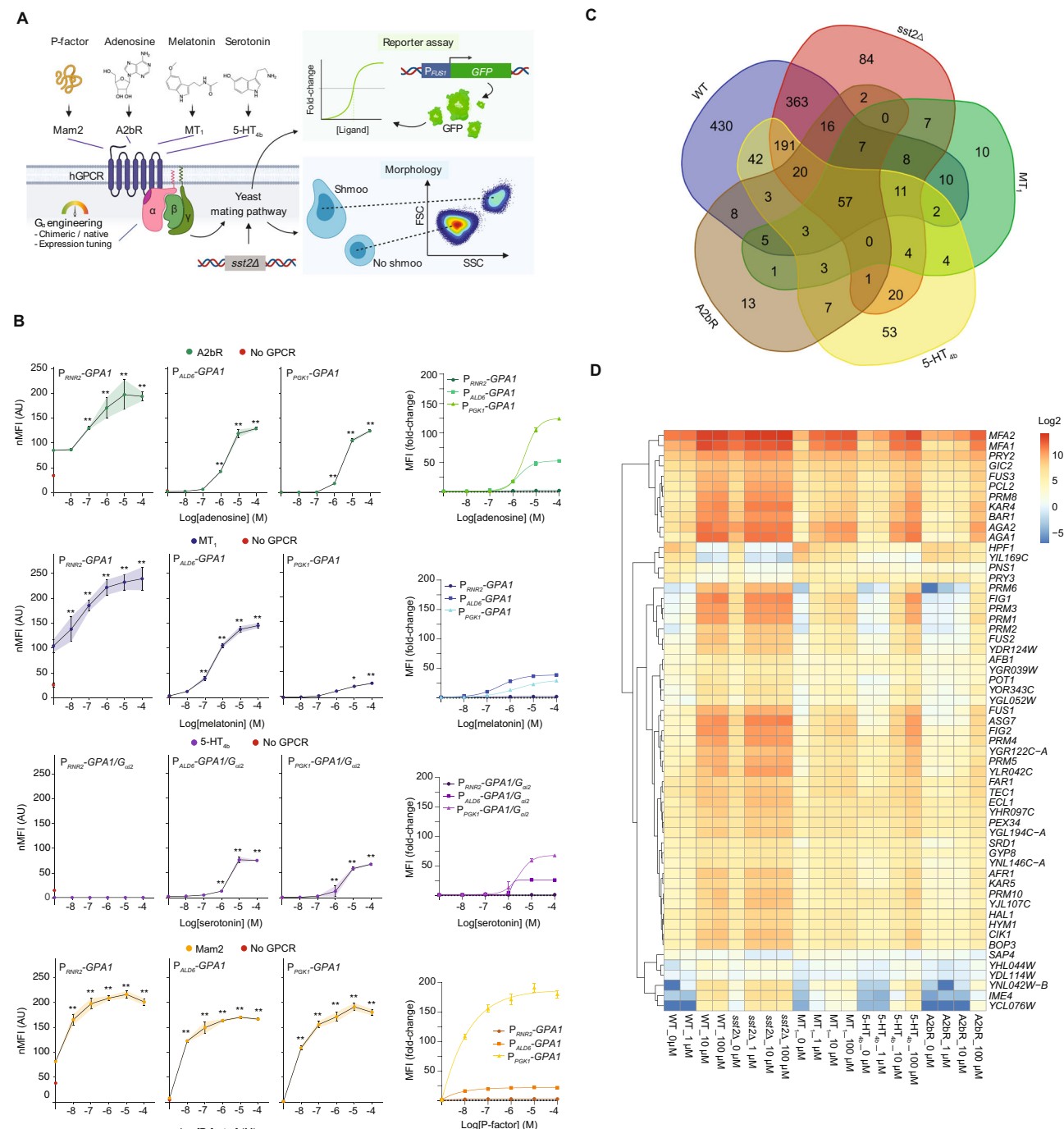

**Fig. 1 | Biosensing yeast characterization and transcriptome analysis.**
**A** Sensitization of *S. cerevisiae* for biosensing with heterologous G protein-coupled receptor (hGPCR)-signaling via its mating pathway. *S. cerevisiae* strains are genetic mutants for *sst2Δ* to diminish inhibition of signal transduction mediated by $G_{\beta\gamma}$ subunits. $G_{\alpha}$ expression tuning controls background and ligand-activated signal transduction schematized as fold change of GFP expressed from the $P_{FUS1}$-GFP reporter ("Reporter assay"). The yeast shmoo phenotype ("Morphology"), following from mating pathway activation, can be observed as increased forward-scatter (FSC) and side-scatter (SSC) with flow cytometry. **B** Normalized median fluorescence intensities (nMFI, left) and MFI fold-changes (right) for biosensing yeast strains expressing A2bR, $MT_1$, $5\text{-}HT_{4b}$, or Mam2 (CPK153-161 and CPK165-167) are shown for low ($P_{RNR2}$), medium ($P_{ALD6}$), and strong ($P_{PGK1}$) $G_{\alpha}$ subunit expression with or without cognate ligands in dosage range 0–100 μM shown as log[M]. Parental strains without GPCRs or ligand supplementation (CPK131-136) are shown as red circles (left). nMFI is normalized for all strains to the $P_{PGK1}$-$G_{\alpha}$ strain without ligand

supplementation within each hGPCR design (left), and fold change is determined relative to the background with no supplementation for each individual strain. Data represent means and standard deviations from three biological replicates, and curves were fitted using GraphPad Prism variable slope (four parameter) nonlinear regression. Means and standard deviations represent three biological replicates. Statistical significance was determined by one-way ANOVA with Dunnett's multiple comparison test in GraphPad Prism (*$p \le 0.05$, **$p \le 0.01$), in relation to the 0 μM state for each strain design. **C** Venn diagram illustrating transcriptome analysis results by depicting overlapping and unique differentially expressed genes (DEGs) between biosensing yeast strains expressing either A2bR (CPK331), $MT_1$ (CPK139), or $5\text{-}HT_{4b}$ (CPK152), vs wild-type (CEN.PK2-1C, WT) and *sst2Δ* sensitized yeast (CPK2) across a gradient of cognate ligand concentrations (0–100 μM). **D** Heatmap showing relative expression for 57 DEGs identified from transcriptome analysis that overlap between all strains and conditions. Source data are provided as a Source Data file.

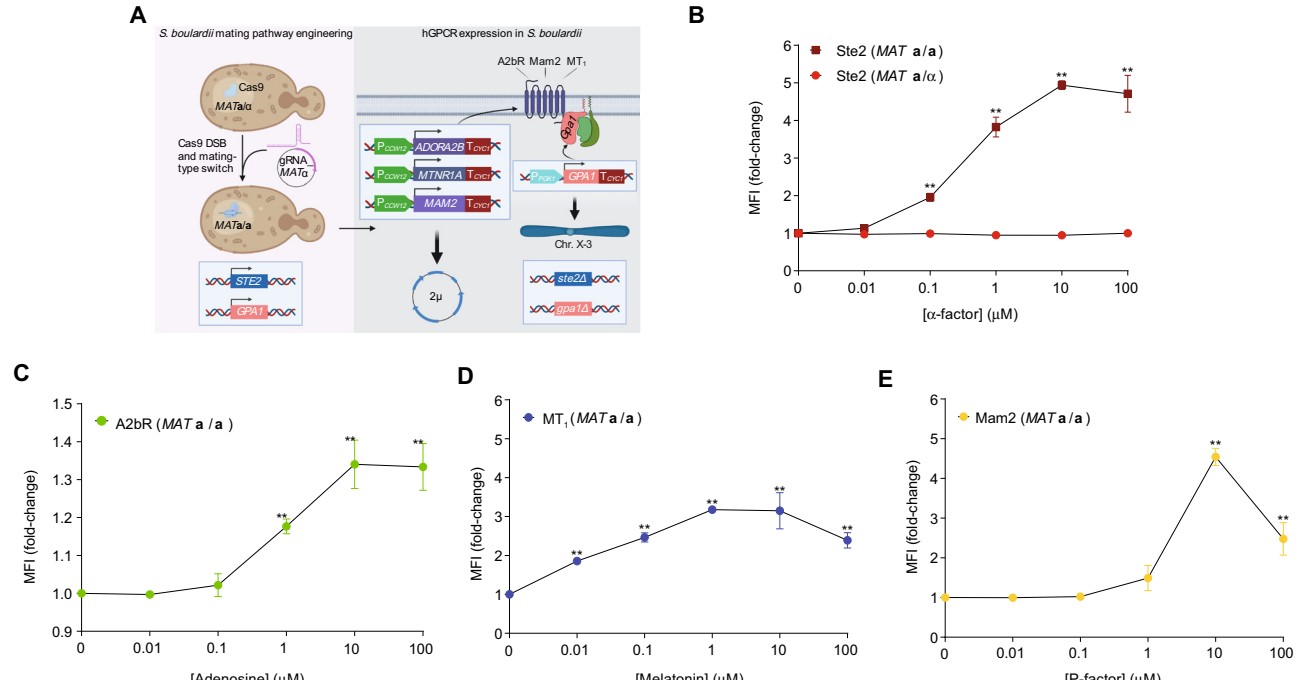

**Fig. 2 | One single genetic edit enables GPCR-signaling in probiotic yeast.**
**A** Homozygous mating-type switch in $MAT\text{a}/\alpha$ diploid cells with CRISPR/Cas9 (left). Cas9 is pre-expressed in *S. boulardii* prior to transformation with a single gRNA plasmid that targets $MAT\alpha$ in the genome for double-strand break (DSB). The endogenous $MAT\text{a}$ copy templates Cas9-induced double-strand break repair (not shown) and converts the cell into a $MAT\text{a}/\text{a}$ diploid. $G_\alpha$ expression is engineered from the *PGK1* promoter, and heterologous GPCRs (hGPCRs) are expressed from 2 μ plasmids (right). **B–E** Median fluorescence intensities (MFI) of $P_{FUS1}$-*GFP* reporter expression from plasmid pDAM194 following incubation with cognate ligands in dosages 0–100 μM. **B**. SB14 ($MAT\text{a}/\alpha$, left) and SB17 ($MAT\text{a}/\text{a}$, right) were incubated with yeast pheromone (α-factor). **C–E** Cognate ligand sensing in hGPCR-signaling $MAT\text{a}/\text{a}$ cells for **C** A2bR sensing adenosine (SB48), **D** MT₁ sensing melatonin (SB49), and **E** Mam2 sensing P-factor (SB50). Means and standard deviations represent three biological replicates. Statistical significance was determined using two-way analysis of variance (ANOVA) with Tukey's multiple comparison test (**B**) or one-way ANOVA with Dunnett's multiple comparison test in GraphPad Prism (**$p \le 0.01$) (**C–E**). Source data are provided as a Source Data file.

yeast. For several decades, the yeast *S. boulardii* has been used to treat and prevent gastrointestinal disorders[26]. However, in the diploid *S. boulardii*, hGPCR-signaling has never been reported, even though this could have major implications to advance the engineering of yeast therapeutics[13]. We hypothesized that the native pheromone GPCRs in *S. boulardii* do not signal because it exists in the diploid $MAT\text{a}/\alpha$ state[27]. Indeed, engineered $MAT\text{a}/\text{a}$ *S. boulardii* can produce viable offspring with *S. cerevisiae*[28], and it was realized decades ago that homozygous *Saccharomyces* $MAT\text{a}/\text{a}$ and $MAT\alpha/\alpha$ diploids are able to mate[29], altogether indicative of pheromone signaling in such diploids.

Thus, we first devised a sniper approach to make homozygous $MAT\text{a}/\text{a}$ diploid *S. boulardii* by targeting the $MAT\alpha$ region for a double-strand break with CRISPR/Cas9[30], in a strategy mimicking native yeast mating-type switching[9] (Fig. 2A). The guide RNA that we used for this procedure targeted both $MAT\alpha$ and HML, which contains a copy of the α genes required for the mating-type switch. Therefore, the activity of *S. boulardii*'s HO endonuclease would not cause reversion to the $MAT\text{a}/\alpha$ state. Following the introduction of the $P_{FUS1}$-*GFP* reporter to these strains, we observed a fivefold Ste2-mediated increased reporter output in the presence of α-factor, while the parental $MAT\text{a}/\alpha$ strain did not respond to any concentrations of α-factor (Fig. 2B).

To equip *S. boulardii* with the design principles for hGPCR-signaling that we learned from *S. cerevisiae* (Fig. 1A, B and Supplementary Fig. 1A), we engineered *S. boulardii* $MAT\text{a}/\text{a}$ strains with $P_{PGK1}$-*GPA1*. We then deleted *STE2* before introducing the genes for hGPCRs A2bR, MT₁, or Mam2, to test reporter output from $P_{FUS1}$-*GFP* following incubation with increasing concentrations of cognate ligands. Here, we observed biosensing in all three probiotic strain designs (Fig. 2C–E). Specifically, A2bR-expressing $MAT\text{a}/\text{a}$ strains showed a modest, but significant, increase in reporter output from 10 μM adenosine

stimulation (1.3-fold) (Fig. 2C). Likewise, reporter output from MT₁- or Mam2-expressing strains increased after stimulation with melatonin (threefold) or P-factor (4.5-fold), respectively (Fig. 2D, E).

As in our biosensing *S. cerevisiae* strain designs (Fig. 1A and Supplementary Fig. 1A), we next deleted the negative mating pathway regulator, *SST2*, to increase reporter output (Supplementary Fig. 2A–C). However, this deletion reduced reporter outputs for all strains to modest maximum fold-changes of 1.5–2.0 fold upon stimulation with their cognate ligands (Supplementary Fig. 2A–C). As a further attempt to increase mating pathway activation, we replaced the strong $P_{PGK1}$ promoter driving expression of *GPA1* with the weaker $P_{RNR2}$. Here, we observed >5-fold increased reporter output from 100 μM α-factor stimulation in $MAT\text{a}/\text{a}$ *SST2* strains (Supplementary Fig. 2D). In contrast, in $MAT\text{a}/\text{a}$ *sst2Δ* strains, $P_{RNR2}$-*GPA1* completely nullified mating pathway signaling upon α-factor stimulation, while $P_{PGK1}$-*GPA1* strain designs reduced background signaling and displayed 7- to 34-fold increased reporter output from α-factor stimulation (Supplementary Fig. 2D).

Taken together, pending simple mating-type switching, we have shown that biosensing strain designs from the model yeast *S. cerevisiae* are functional in probiotic *S. boulardii*, albeit necessitating further exploration of G protein balancing and hGPCR-signaling in this organism.

### Morphologies of yeasts with hGPCR-activated mating pathways

Chemotropism and navigation along chemical gradients are universal traits of living organisms[31]. In *S. cerevisiae*, shmooing is a hallmark of chemotropism and inherent to mating pathway activation and successful cell-cell conjugation[3]. In search of optimal parameters for hGPCR-mediated activation of the mating pathway, we initially tested

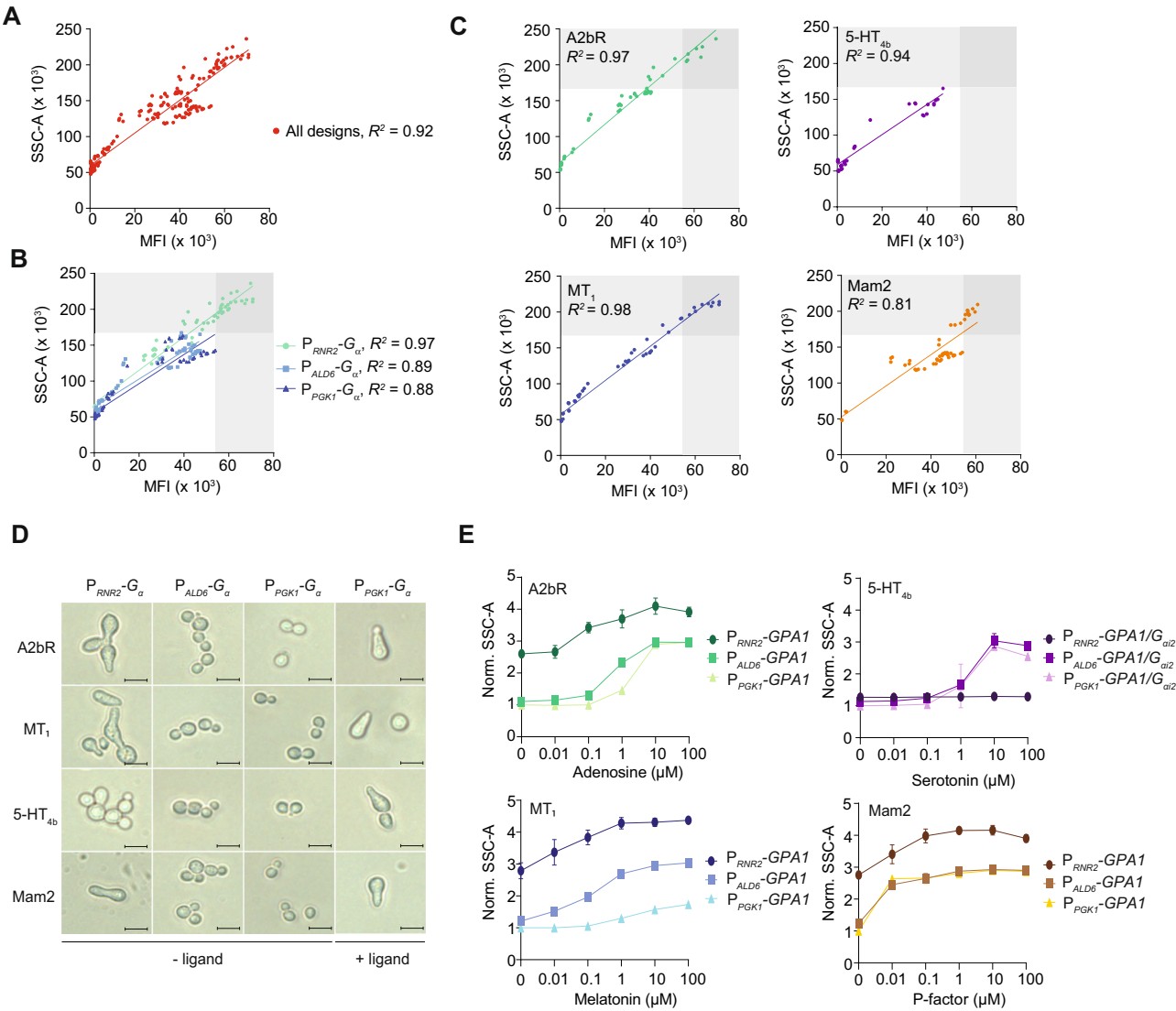

**Fig. 3 | Mating pathway activation and changed morphology correlate.**
**A**–**C** Increased side-scatter area (SSC-A) correlates with median fluorescence intensities (MFI) from $P_{FUS1}$-$GFP$ reporter expression across cognate ligand dosages ranging 0–100 μM in A2bR, MT₁, 5-HT₄ᵦ, and Mam2 biosensing strains of *S. cerevisiae* (CPK153-161 and CPK165-167). Linear regression was performed using Graphpad Prism, and R² is reported for each analysis. **A** All data (*n* = 216). **B** Highlights and correlation of each $G_\alpha$ expression design in all data (*n* = 72 × 3). **C** Individual representation of correlation for all hGPCR designs (*n* = 54 × 4).

**D** Representative morphologies in *S. cerevisiae* strains from panels **A**–**C** in the absence (−ligand) or presence of cognate ligands (+ligand, 100 μM) for A2bR, MT₁, and 5-HT₄ᵦ, or Mam2 (1 μM). The experiment was repeated more than three times. Size bars illustrate 10 μm. **E** Increasing SSC-A for each strain design in panels **A**–**C** across 0–100 μM cognate ligand supplementation as indicated. SSC-A is normalized for all strains to the $P_{PGK1}$-$G_\alpha$ strain without ligand supplementation within each hGPCR design. Data represent means and standard deviations from three biological replicates. Source data are provided as a Source Data file.

if hGPCR-signaling yeast strains could shmoo across increasing concentrations of cognate ligands. Here, we defined shmoos from side-scatter (SSC-A) quantified by flow cytometry, which was validated using microscopy as previously described[5,32]. Multivariate analysis across all biosensing *S. cerevisiae* strain designs revealed a greater correlation between the $P_{FUS1}$-$GFP$ reporter and SSC-A ($R^2$ = 0.92) than with forward-scatter (FSC-A) ($R^2$ = 0.81) (Fig. 3A), previously used for shmoo quantification[33]. Thus, the complex shmooing response, spanning >4 orders of magnitude of ligand concentrations, can be directly correlated to reporter fluorescence intensity.

We noticed that individual design parameters were important determinants of the observed correlation between shmooing and reporter fluorescence. Specifically, $P_{RNR2}$-$G_\alpha$ designs displayed greater shmoo-fluorescence correlations compared to $P_{ALD6}$-$G_\alpha$ and $P_{PGK1}$-$G_\alpha$ designs ($R^2$ = 0.97 vs $R^2$ = 0.89 and $R^2$ = 0.88, respectively) (Fig. 3B), although A2bR and MT₁ designs exhibited strong

correlations ($R^2$ = 0.97) regardless of $G_\alpha$ expression design (Fig. 3C). Likewise, and in agreement with *FUS1* being identified as a core DEG from our RNA-seq study (Fig. 1C, D), the $P_{PGK1}$-$G_\alpha$ strain designs with no or low background activation of the mating pathway (Fig. 1B) also showed strong correlations ($R^2$ = 0.92–0.99) between shmooing and reporter fluorescence for all biosensing strains (Supplementary Fig. 3A). Microscopy confirmed the shmoo morphology interpreted from SSC-A (Fig. 3D, E, Supplementary Fig. 3B, D). Lastly, increased SSC-A was also apparent for the $P_{PGK1}$-$G_\alpha$ strains composed in *MAT***a**/**a** *S. boulardii* across gradients of cognate ligands (Supplementary Fig. 3C).

Altogether, and in agreement with classical reporter assays[6,14], this data shows that mating pathway activation of hGPCR-expressing biosensing yeasts can be tightly controlled across large ligand gradients, depending on biosensing strain design, and correlates well with morphological changes indispensable for mating and cell-cell conjugation.

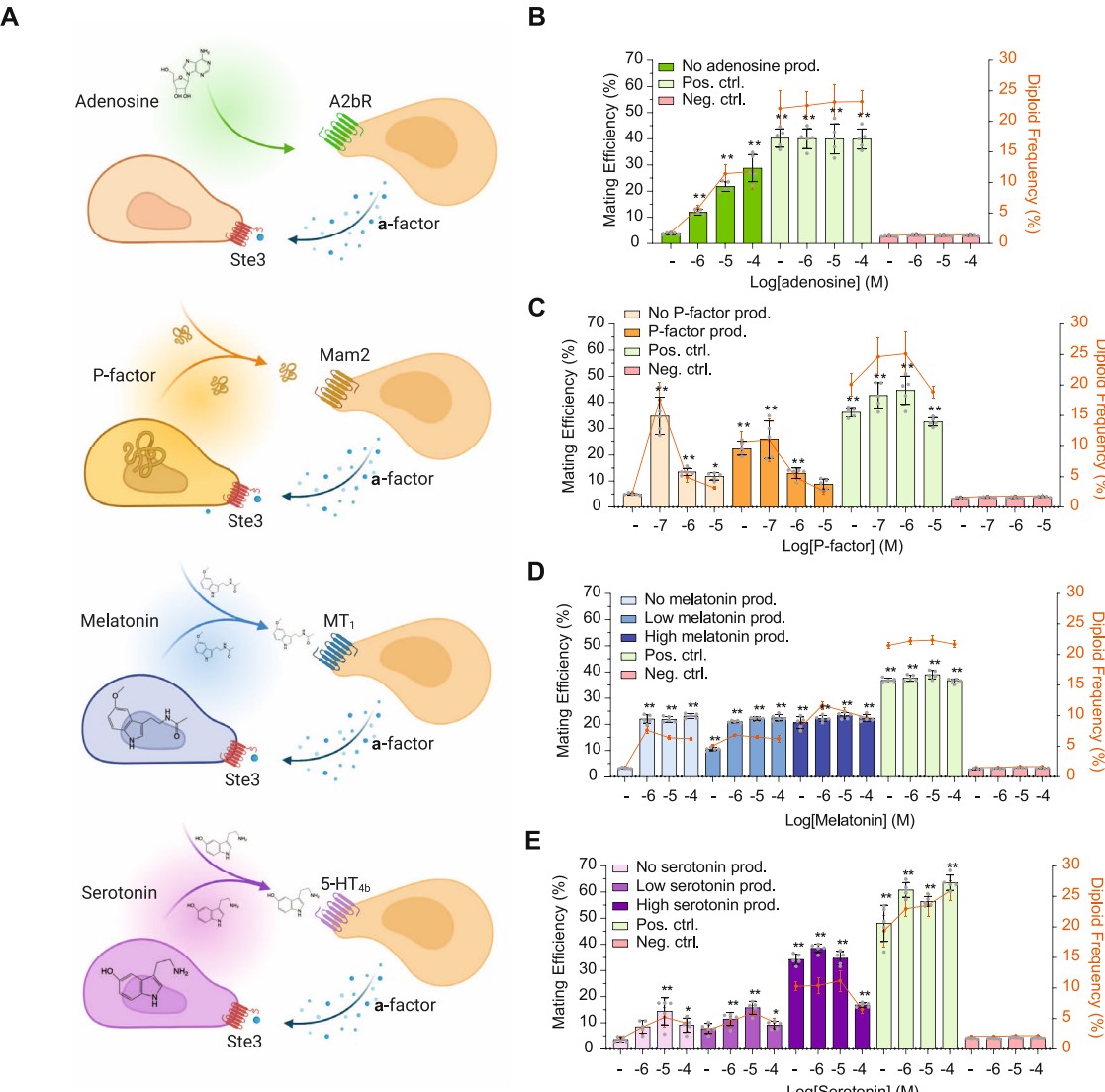

**Fig. 4 | Heterologous GPCRs support semi-synthetic yeast mating. A** Schematic outline showing production, supplementation, and biosensing of melatonin (MT$_1$), serotonin (5-HT$_{4b}$), or P-factor (Mam2) in yeast mating pairs. Adenosine was introduced solely by supplementation as illustrated for sensing by A2bR.
**B–E** Percentage of mating efficiencies (second-axis) and diploid frequencies (third-axis) at increasing cognate ligand supplementation dosages (first-axis, 0–100 μM shown as log[M]) were scored following 5 h co-incubation at 30 °C and 250 RPM in a 96-deep-well plate. Results are presented as means with standard deviations determined from five biological replicates. CPK46xSBY55 were used as a positive control for native mating throughout. CPK121 (*GPA1*) and CPK124 (*GPA1/G$_{ai2}$*) do not contain any GPCRs and were used as negative mating controls with SBY55 in panels **B**–**D** (CPK121) and in panel **E** (CPK124). No ligand supplementation controls are indicated (−). **B** CPK331xSBY55 mating at different supplementation doses of

adenosine. **C** CPK142xSBY55 (no production) and CPK142xSBY155 (P-factor production) mating crosses were investigated with and without additional supplementation of P-factor in 1% DMSO in indicated concentrations. **D** CPK139xSBY55 (no production), CPK139xSBY92 (low melatonin production), and CPK139xSBY139 (high melatonin production) mating crosses were investigated with and without additional supplementation of melatonin in indicated concentrations.
**E** CPK152xSBY55 (no production), CPK152xSBY91 (low serotonin production), and CPK152xSBY138 (high serotonin production) mating crosses were investigated with and without additional supplementation of serotonin in indicated concentrations. Statistical significance was determined for mating efficiencies using two-way analysis of variance (ANOVA) with Dunnett's multiple comparisons test, relative to the condition of no production and no supplementation for each setup, using GraphPad Prism (*$p \le 0.05$, **$p \le 0.01$). Source data are provided as a Source Data file.

## Semi-synthetic mating in hGPCR-signaling yeast

Having confirmed that hGPCR activation evokes shmooing in biosensing yeast strains, we next asked if this activation also translated into mating. In the following, we define mating between haploid *MAT***a** and *MAT*α cells with single-receptor substitutions of *STE2* in *MAT***a** cells as "semi-synthetic mating". As measures of sexual reproduction between engineered haploid cells, we report both population-wise diploid frequencies and calculated mating efficiencies, with the former refering to the proportion of the entire sampled mating culture identified as diploids, while the latter describes how efficiently the given amount of diploids have formed relative to the amount of potential mating events.

To investigate if biosensing strains expressing hGPCRs could support semi-synthetic mating, we initially substituted the wild-type *MAT***a** partner cell with a biosensing yeast strain for P-factor, serotonin, or melatonin to conduct mating trials with a *MAT*α cell that retained its native Ste3 receptor. *MAT*α cells were engineered to produce the synthetic pheromones (P-factor, serotonin, or melatonin) to mimic native paracrine signaling, while the biosensing yeast strains all retained native **a**-factor expression (Fig. 4A). In addition, we included a *MAT*α mating partner without production to investigate if supplementation of cognate ligands would present a viable strategy to control mating as compared to endogenous production. Due to the low

background activation of the mating pathway and morphological changes observed for strain designs with high expression of $G_\alpha$ (Figs. 1B, 3B, D), we focused our attention on hGPCR-expressing strains with $P_{PGK1}$-$G_\alpha$ designs.

We first evaluated the cognate ligand supplementation strategy without production for the adenosine biosensing strain. Here we observed a maximum mating efficiency of 29%, which increased ~8-fold from 0–100 μM adenosine supplementation (Fig. 4B). By comparison, wild-type $MAT\mathbf{a}$ x $MAT\alpha$ (positive control) and receptor-deficient $MAT\mathbf{a}$ x $MAT\mathbf{a}$ (negative control) crosses showed 40 and 3% mating efficiencies, respectively. Similarly, diploid formation from semi-synthetic mating based on supplementation of adenosine reached up to 12% of the screened population, whereas wild-type crosses amounted to 23% (Fig. 4B).

Having confirmed cognate ligand supplementation as a viable strategy for semi-synthetic mating, we next investigated mating efficiencies from endogenous production of synthetic pheromones with and without additional supplementation. Here, the biosensing strain expressing Mam2 crossed with a strain producing P-factor resulted in a mating efficiency of 22% without supplementation compared to 5% in crosses without production or supplementation (Fig. 4C). The same pattern was observed for crosses with strains producing melatonin or serotonin (Fig. 4D, E and Supplementary Data 3), and the resulting diploids also displayed increased SSC-A compared to haploids, the extent of which was related to the hGPCR stimulation applied during mating (Supplementary Fig. 4). Production of synthetic pheromones for crosses with P-factor and melatonin biosensing strains showed similar mating efficiencies compared with those observed for supplementation of cognate ligands (Fig. 4C, D), whereas for the serotonin biosensing strains, high production outperformed the best mating efficiencies observed at 10 μM supplementation alone (34 vs 14%) (Fig. 4E and Supplementary Data 3). Strikingly, supplementation of 100 μM serotonin in the high-production crosses yielded reduced mating efficiency compared to lower concentrations of cognate ligand supplementation (Fig. 4E). This was also observed for P-factor supplementation in excess of 0.1 μM, irrespective of P-factor production (Fig. 4C), indicating a potential mating pathway overstimulation which could also be observed for these strains in the initial mating pathway characterization (Fig. 1B). Likewise, while this pattern was not observed for the dosages tested in this study for the A2bR- and MT_1-expressing strains (Figs. 1B, 4B, D), the positive correlation observed between shmooing and mating efficiencies at ligand concentrations up to 10 μM diluted at higher concentrations of supplemented ligands (Figs. 3E, 4C, E), underscoring that even though shmooing is indispensable for mating and cell-cell conjugation, the size of shmoos are not the sole determinant of mating efficiency.

In sum, these results show that replacing Ste2 with human or fungal GPCRs in biosensing yeast strains enables efficient semi-synthetic mating, and that mating efficiencies do not directly relate to shmoo sizes, yet are controllable by cognate ligand supplementation and synthetic pheromone production.

## Full synthetic and autonomous yeast mating

To further explore the contribution from each hGPCR in the complete absence of native pheromone signaling, we substituted pheromone receptors in both mating partners for hGPCRs. As for semi-synthetic mating trials, we initially supplemented the cognate ligands exogenously, either individually or in combination for each mating pair (Fig. 5A).

Here, supplementation of adenosine and melatonin together revealed synergistic mating efficiencies starting at 1 μM for each ligand. The highest mating efficiency was observed at 100 μM adenosine and melatonin combined (40%), while 10 μM of each ligand yielded the most frequent diploid formation (7.6%) (Fig. 5B and Supplementary Fig. 5). When supplied individually, only melatonin clearly increased

mating efficiencies, corroborating the results obtained in cells without melatonin production (Figs. 4D, 5B). Yet, the mating efficiencies and diploid formation frequencies observed from the supplementation of individual ligands were never as high as for simultaneous stimulation with both cognate ligands.

In another mating cross with adenosine and serotonin biosensing strains, individually supplementing 10 μM adenosine or serotonin resulted in elevated mating efficiencies only with serotonin supplementation (4.5–7.3%) relative to no supplementation (3.6%) (Fig. 5C). Supplementation of both ligands together at 10 μM resulted in the highest mating efficiency (11%) and diploid formation (4.5%) for this mating pair (Fig. 5C). Conversely, in mating pairs expressing $MT_1$ and 5-HT_{4b}, we observed no increased mating efficiency from only supplementing serotonin, whereas melatonin supplementation alone again resulted in elevated mating efficiencies (3.6–6.6%) (Fig. 5D). Importantly, and just as for the results obtained with supplementation of adenosine and melatonin in synthetic mating pairs expressing A2bR or $MT_1$ (Fig. 5B), supplementation of both ligands at 10 μM gave the highest mating efficiency (10%) (Fig. 5D). Furthermore, albeit with a different impact from each hGPCR in synthetic mating pairs, decreased diploid frequencies were apparent at the highest supplementation levels across all three biosensing yeast mating pairs (Fig. 5B–D).

Although mating efficiency for strains that sense melatonin and adenosine reached 40% (Fig. 5B), mating efficiency only reached ~10% at best for strains sensing serotonin and adenosine or serotonin and melatonin (Fig. 5C, D). As noticed earlier, biosensing strains elicit different levels of responses depending on the hGPCR they express (Figs. 1B, 3, 4), and and it, therefore, appeared likely that tuning the ratio of the mating pairs could improve the mating efficiency. By supplementation of both ligands (10 and 100 μM) we discovered that 10x ratio tuning improved mating efficiency by ~3-fold for the serotonin and adenosine mating pair (Fig. 5E), and ~2-fold for the serotonin and melatonin mating pair (Fig. 5F). In this way, ratio tuning can be used to increase mating efficiency for synthetic mating.

Finally, we investigated full autonomous mating between biosensing yeasts expressing Mam2 or 5-HT_{4b} with endogenous production of synthetic pheromones (P-factor and serotonin) (Fig. 5G). Here, serotonin was produced continuously from one strain (SBY175) while P-factor was expressed in a partner strain (CPK508-511), either constitutively from $P_{TEF1}$ or dynamically regulated from one of the pheromone-responsive promoters $P_{AGA2}$, $P_{MFA1}$, or $P_{FUS1}$ (Fig. 1D and Supplementary Data 1) during 5-HT_{4b} signaling. Once again, we applied ratio tuning to increase mating efficiency and found that a 10× surplus of SBY175 relative to each CPK strain gave the highest mating efficiency in every trial ($P_{TEF1}$: 9%; $P_{AGA2}$: 12%; $P_{MFA1}$: 15%; $P_{FUS1}$: 26%) (Fig. 5G). Thus, the highest mating efficiency resulted from the engineering of dynamically regulated P-factor expression from $P_{FUS1}$, which, in combination with ratio tuning, gave ~3.5-fold improvement over the negative control lacking both GPCRs and paracrine signaling.

Altogether, our results show that both pheromone receptors in a yeast mating pair can be functionally replaced for hGPCRs, and that the mating efficiency and diploid frequencies can be controlled by the dosage of their cognate ligand pairs, and to a lower extent by only supplementing single synthetic pheromones. Most importantly, our data also demonstrate full paracrine signaling in a fully autonomous manner, as demonstrated by mating between 5-HT_{4b}- and Mam2-expressing haploid cells producing the synthetic pheromones P-factor and serotonin, respectively.

## Discussion

This study covers systemic transcriptional perturbations upon hGPCR-signaling and mating pathway activation in yeast. It further demonstrates successful cell differentiation and autonomous synthetic sexual reproduction in a model eukaryote, as well as mating pathway activation in probiotic yeast, *S. boulardii*. It offers a new resource for

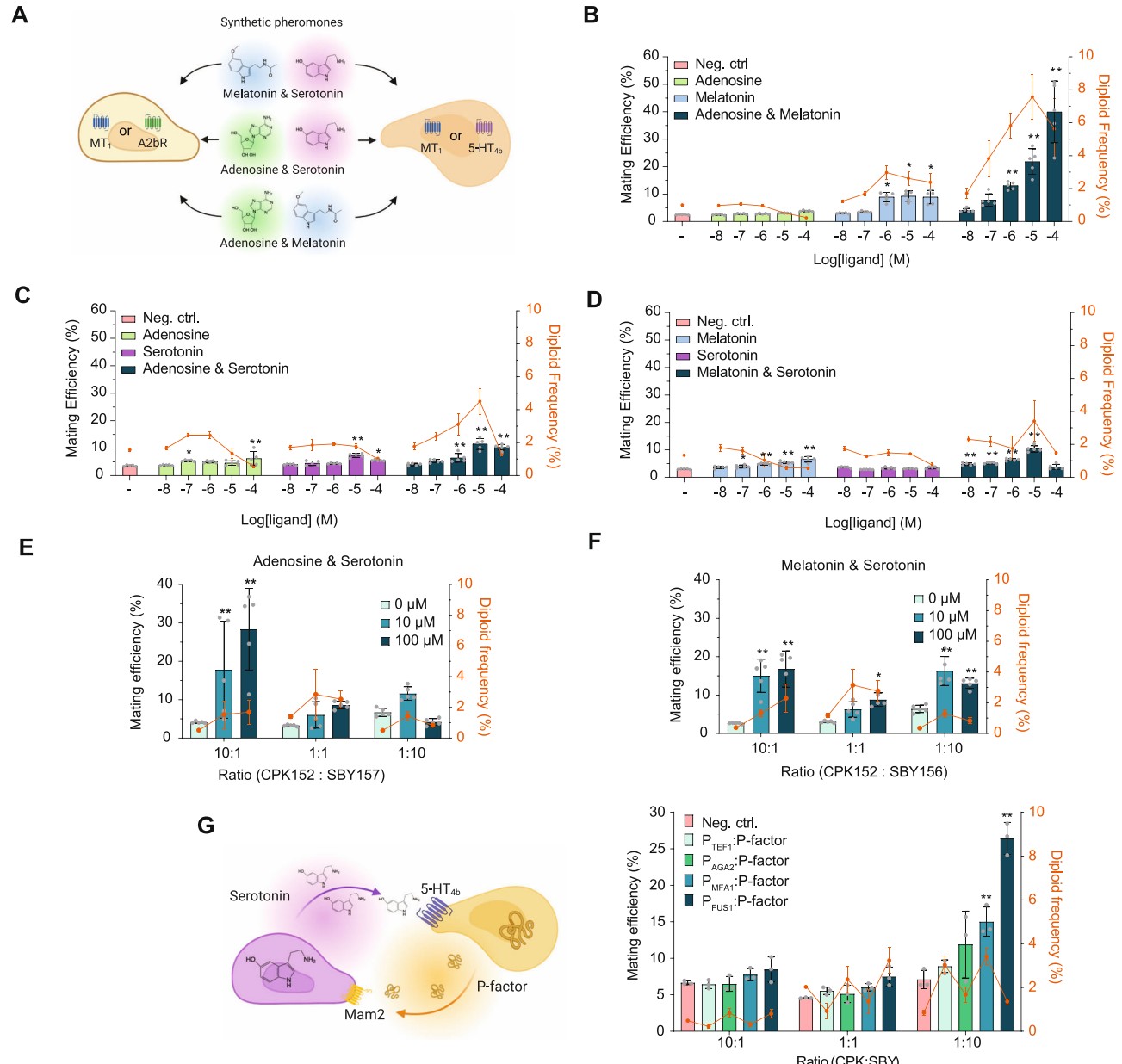

**Fig. 5 | Heterologous GPCRs support full synthetic and autonomous yeast mating. A** Individual or combined synthetic pheromone (cognate ligand) supplementation and biosensing of adenosine (A2bR), melatonin (MT₁), and serotonin (5-HT₄b) in yeast mating pairs. **B–D** Percentage of mating efficiencies (second-axis) and diploid frequencies (third-axis) following supplementation of one or both synthetic pheromones at increasing dosages (first-axis, 0–100 μM shown as log[M]) were scored following 5 h co-incubation at 30 °C and 250 RPM in a 96-deep-well plate. Results were presented as means with standard deviations determined from five biological replicates. Negative controls without ligand supplementation are indicated (−). Mating crosses were done with **B** adenosine or melatonin supplementation alone or in combination for CPK139xSBY157, **C** adenosine or serotonin alone or in combination for CPK152xSBY157, and **D** melatonin or serotonin alone or in combination for CPK152xSBY156. **E, F** Strain ratios of 10:1, 1:1, and 1:10 were investigated for mating pairs CPK152xSBY157 (**E**) and CPK152xSBY156 (**F**) in 5 h

co-incubations with 0, 10, or 100 μM adenosine and serotonin (**E**) or melatonin and serotonin (**F**). Data represent means and standard deviations from at least three biological replicates. **G** Paracrine signaling from biosensing of serotonin and P-factor production in autonomous yeast mating pairs. SBY175, expressing the GPCR Mam2 and producing serotonin, was crossed in ratios 1:10, 1:1, and 10:1 with strains expressing the GPCR 5-HT₄b and producing P-factor constitutively from P_TEF1 (CPK508), or in dynamically regulated response to serotonin from the pheromone-inducible promoters P_AGA2, P_MFA1, or P_FUS1 (CPK509-511). CPK124xSBY172 with no GPCRs or ligand production were used as negative controls. Bars represent averages from three biological replicates. Statistical significance was determined for mating efficiencies using one-way (**B–F**) and two-way (**G**) analysis of variance (ANOVA) with Dunnett's multiple comparisons test, relative to the negative control in each setup, using GraphPad Prism (*$p \leq 0.05$, **$p \leq 0.01$). Source data are provided as a Source Data file.

exploring the control of biosensing and cellular morphology by the use of regulatory elements in yeast expressing either native or heterologous GPCRs and $G_\alpha$ proteins. Taken together, this study highlights crucial design parameters and environmental conditions to tune mating pathway activation, cellular differentiation, and synthetic mating.

With respect to recommended strain designs for mating pathway activation, we consider "coupling-shift" as the first feature to assess. By design, weak reporter expression is thought to arise from hGPCR-$G_\alpha$ coupling, in which the hGPCR exerts GEF function on spontaneously arising nucleotide-free $G_\alpha$ subunits that leads to ligand-independent signal transduction by pushing the equilibrium towards $G_{\beta\gamma}$ release in

sensitive biosensing yeast strains. Thus we envision future studies to leverage coupling-shifts for cheap ligand-free pre-screens to accelerate the identification of signaling-competent hGPCR-$G_\alpha$ pairs prior to further studies, e.g. deorphanization. Likewise, tuning of $G_\alpha$ expression enables high-throughput assessment of nonlinear dosage-dependent shmoo sizes, thereby helping to determine optimal balancing of paracrine signaling for successful synthetic mating. Specifically, based on this study, we recommend the expression of $G_\alpha$ using a weak promoter for assessing hGPCR-coupling to the mating pathway, and a strong promoter for initial testing of synthetic mating as this design demonstrated low basal activity, high dynamic output ranges (up to 200-fold), as well as relatively large changes in shmoo sizes (Figs. 1B, 3E).

However, the engineering of sexual reproduction at large needs careful consideration beyond the choice of hGPCR, the expression level of $G_\alpha$, and types of synthetic pheromones towards successful paracrine signaling, coordinated chemotropism, and mating. For instance, as evidenced by declining synthetic mating efficiencies and diploid formation from supplementation of synthetic pheromones at high concentrations (Figs. 4C, E, 5C, D), increased reporter expression during hGPCR-signaling does not necessarily capture mating pathway overstimulation (Figs. 1B, 3E). Also, even though shmoo sizes and mating efficiencies both declined at high dosages for the $P_{PGK1}$-$G_\alpha$ design with 5-HT$_{4b}$ (Fig. 4E), the same correlation was not apparent for the almost identical Mam2 strain design (Figs. 3E, 4C). Extending from these observations, the establishment of optimal pheromone gradients for efficient cell-cell communication and mating using synthetic pheromones should greatly benefit from the identification of new barrier proteins, or metabolic enzymes, analogous to pheromone-binding/-cleaving *ABF1* and *BAR1* from *MAT*α and *MAT***a** cells, respectively[34,35]. Most importantly, simple ratio tuning between synthetic mating pairs enabled 2- and 3-fold mating efficiency improvements over 1:1 ratios at supplementation with both ligands (Fig. 5E, F) and up to 3.5-fold mating efficiency improvement for autonomous mating pairs when serotonin was endogenously produced and dynamically regulated pheromone-inducible P-factor expression was introduced (Fig. 5G).

Lastly, this study is, to the best of our knowledge, the first demonstration in the probiotic *S. boulardii* of mating pathway activation controlled by signaling from hGPCRs (Fig. 2), yet further studies are encouraged to elucidate the apparent different role that the negative regulator of mating pathway activation, *SST2*, plays in this conspecific yeast (Supplementary Fig. 2A–D) compared to *S. cerevisiae*. Indeed, from understanding the regulatory mechanisms of the mating pathway, and taking advantage of *S. boulardii*'s growth, survival, and residence time in the gastrointestinal tract as compared to *S. cerevisiae*[36], we envision that probiotic *S. boulardii* can now be engineered to dynamically respond to disease markers for timely and accurate secretion of therapeutic compounds, and as such adds an important capability to its already established therapeutic potential. Extending from this, we furthermore foresee this study to foster the development of growth-based high-throughput screens of hGPCR-mediated chemotropism within environmental engineering, MAPK pathway signaling, drug discovery for therapeutic purposes, and enzyme optimization for metabolic engineering and biotechnology.

## Methods
### Molecular cloning
**Standard techniques and materials.** USER cloning[37] was used to construct plasmids, unless otherwise specified, and the EasyClone MarkerFree system with integration plasmids and gRNA plasmids compatible with CRISPR/Cas9[38] were used throughout. USER-compatible vectors were treated with FastDigest *Sfa*AI (Thermo Fisher Scientific) and Nb.*Bsm*I (NEB) prior to ligation, and USER-compatible fragments were amplified with Phusion U Hot Start PCR Master Mix (Thermo Fisher Scientific) to read through uracil overhangs contained in oligos. Strains, plasmids, oligos, gene blocks, and heterologous GPCR accession IDs are listed in Supplementary Data 4–8.

**Codon optimization of gene blocks.** Gene blocks gDAM5 and gDAM8-14 were codon optimized for *Saccharomyces cerevisiae* with the IDT Codon Optimization tool. All oligos and gene blocks were purchased from IDT, and coding sequences were immediately preceded by a transcriptional enhancer sequence (AAAACA).

**gRNA plasmids.** Single gRNAs were introduced to existing gRNA plasmids by inverse PCR and blunt-end ligation with T4 DNA ligase (NEB). Oligos DAM1-4 and DAM6 were used individually with DAM594 to amplify pCfB3050 making up plasmids pDAM1-4 and pDAM6. pDAM7 and pDAM8 were made by amplifying the gRNA cassettes from pDAM1 and pDAM2 with, oligos TJOS-62 and TJOS-65 and assembled into pEDJ400 and pEDJ437[39], respectively. Oligos DAM218 + DAM594 and DAM219 + DAM594 amplified pDAM1 to give plasmids pDAM77 and pDAM78, DAM485 + DAM594 amplified pDAM8 to make pDAM182, and DAM335 + DAM594 amplified pDAM7 to make pDAM146. gRNA cassettes from pDAM77 and pDAM78 were amplified with TJOS-62 + TJOS-66 and TJOS-63 + TJOS-65[40], respectively, and assembled into pEDJ437 to give pDAM82. Plasmids ID6911 and PL_12_I3 were made from inverse amplification of pDAM7 with oligos JZ1 + DAM594 and MAD1 + DAM594, respectively. pDAM135 was created by amplification of the PL_12_I3 cassette using TJOS-62 + TJOS-65 and assembly into pEDJ437. TJOS-62 + TJOS-66, TJOS-63 + TJOS-67, TJOS-64 + TJOS-65 amplified pCfB3042, pCfB3045, and pCfB3049 gRNAs, respectively, and were assembled together into pTAJAK-71[40] to make the triple gRNA plasmid pDAM22. pDD110 was made with *Bsa*I golden gate cloning by the assembly of the gene block DDgb008 and YTK003 (Con2), YTK010 (pCCW12),YTK036 (Cas9), DDgb008 (pSNR52, GFP dropout, sgRNA, tSUP4), YTK055 (tENO2), YTK071 (Con5), YTK078 (*NatR*), YTK081 (CEN6/ARS4), and YTK084 (*KanR*-ColE1) according to the protocol for the YTK toolkit[41]. gRNA sequences were ordered as annealed oligos (DDpr262 + DDpr263 for gRNA_*URA3* and DDpr264 + DDpr265 for gRNA_*HIS3*) with 4 bp overhangs, and with golden gate, cloning entered pDD110 to make pDD111 and pDD115.

**$G_\alpha$ plasmids construction.** Promoters $P_{RNR2}$, $P_{ALD6}$, and $P_{PGK1}$ were amplified from yeast genomic DNA (gDNA) with oligos EDJ134 + DAM116, DAM117 + DAM118, and DAM58 + EDJ318, respectively. *GPA1* was amplified with oligos DAM81 + DAM82 from CEN.PK2-1C gDNA, and *GPA1*/$G_{\alpha i2}$ from gene block gDAM1 with oligos DAM81 + DAM83. $P_{PGK1}$, $P_{RNR2}$, or $P_{ALD6}$ were then assembled with *GPA1* or *GPA1*/$G_{\alpha i2}$ into pRS415U to give plasmids pDAM30, pDAM32, pDAM47, pDAM49, pDAM50, and pDAM52 respectively.

**hGPCR plasmid constructions.** The promoter $P_{CCW12}$ was amplified from gDNA with oligos DAM191 + DAM192, and the terminator $T_{CYC1}$ with oligos DAM59 + DAM193. hGPCRs A2bR (*ADORA2B*), MT$_1$ (*MTNR1A*), Mam2[8], and 5-HT$_{4b}$ (*HTR4*)[21] were amplified from gene blocks gDAM2-5 with oligos DAM63 + DAM64, DAM65 + DAM66, DAM67 + DAM68, and DAM160 + DAM162, respectively, and each fragment was USER assembled with $P_{CCW12}$ and $T_{CYC1}$ into pEDJ437 to give plasmids pDAM71, pDAM72, pDAM74, and pDAM76, respectively. Plasmids pDAM216–223 expressing hGPCRs hβ$_2$-AR (*ADRB2*)[5], CXCR4[42], GLP-1R[24], CaSte2, FgSte2 or ZtSte2[7], and TrSte2 (Uniprot ID: A0A022VRI2) or MsSte3 (Uniprot ID: A0A1M8A1X3), individually encoded in gDAM7-14, were amplified with oligos DAM335-DAM350, respectively, and assembled as before with $P_{CCW12}$ and $T_{CYC1}$ into pEDJ437.

**Plasmids for synthetic pheromone production—serotonin, melatonin, and P-factor.** All expression cassette plasmids used as templates for PCR have been previously described[43]. pDAM23 ($T_{ADH1}$-$RnPTS < $-$P_{TEF1}$-$P_{PGK1} > RnSPR$-$T_{CYC1}$) was made from amplification of pCfB1251 using oligos 350 + 389 and assembly into pCfB3035. pDAM24 ($T_{ADH1}$-$PaPCBD1 < $-$P_{TEF1}$-$P_{PGK1} > RnDHPR$-$T_{CYC1}$) was made from amplification of pCfB1248 using oligos 2149 + 2153 and assembly into pCfB3040. To generate pDAM25 ($T_{ADH1}$-$HsASMT < $-$P_{TEF1}$-$P_{PGK1} > BtAANAT$-$T_{CYC1}$), *BtAANAT* was amplified from pCfB2628 using oligos 1761 + 1762, *HsASMT* from pCfB1252 using oligos 2254 + 2255, as well as bidirectional joint promoters $P_{TEF1}$-$P_{PGK1}$ from pCfB2628 using oligos 5 + 8, and subsequently all parts were assembled into pCfB2899. pCfB9221 contains $T_{ADH1}$-$HsDDC < $-$P_{TDH3}$-$P_{TEF1} > SmTPH$-$T_{CYC1}$.

**Other plasmids.** T>he Cas9 expression cassette from pEDJ391 was amplified with oligos 1564 + EDJ325 and USER assembled into pRS416U to make plasmid pDAM215. *mKate2* was amplified from pYR11[44] with oligos DAM288 + DAM289 and assembled with $P_{TEF1}$ amplified from gDNA with oligos 1564 + 1565 to make plasmid pDAM122. pDAM123 and pDAM236-238 were made by assembling fragments from gDAM15 (α-leader secretion signal and P-factor) amplified with DAM76 + DAM78 and $P_{TEF1}$ (1564 + 1565), $P_{FUS1}$ (DAM596 + DAM655), $P_{Aga2}$ (DAM598 + DAM656), or $P_{MFA1}$ (DAM657 + DAM658) into pCfB2899. Plasmid pJV452 was constructed by Gibson (NEB) assembly of pEDJ400 linearized with oligos JV498 + JV499, which excluded the 2μ element, and CEN/ARS from plasmid DD118[36] amplified with oligos JV500 + JV501. Plasmid pDAM194 containing $P_{FUS1}$-$yEGFP$-$T_{FUS1}$ was made by amplification of gDNA from strain CPK16 with oligos JV502 + JV503, which was then assembled into plasmid pJV452.

### Engineering of *S. cerevisiae*
Strain CEN.PK2-1C and BY4741 (EUROSCARF) were transformed with pEDJ391[39] to express Cas9 (CPK1 and SBY1). About 1–2 μg for each plasmid or fragment of DNA was used in all chemical transformations of *S. cerevisiae*.

**Baseline *S. cerevisiae* strains.** CPK1 and SBY1 were consecutively genetically deleted for *SST2*, *STE2*, and *STE3* with gRNA plasmids pDAM4, pDAM7, and pDAM8 and homology templates amplified as two overlapping fragments from gDNA with oligos DAM19-22 for *sst2Δ*, DAM7-10 for *ste2Δ*, and DAM13-16 for *ste3Δ* to give strains CPK2-4 and SBY2-4, respectively. In addition, a landing-pad-based strain was engineered using the same approach but with amplification of landing-pads from yWS677[8] with oligos DAM19 + DAM22, DAM8 + DAM10, and DAM13 + DAM16 for *sst2Δ; ste2Δ; ste3Δ*, respectively, resulting in platform strain CPK88. Next, the pheromone-responsive *FUS1* promoter and terminator were amplified with oligos DAM53-56. Both fragments included overlapping homology to a yeast-enhanced green fluorescent protein (*yEGFP*) amplified from the gene block gDAM6 with oligos DAM39 + DAM40. Co-transformation of these three fragments and the gRNA plasmid pDAM3 into strains CPK1, CPK4, and SBY4 replaced the native *FUS1* open reading frame with *yEGFP* to give CPK86, CPK16, and SBY16, respectively. CPK86 was genetically deleted for *SST2* as described for CPK88 to make strain CPK503.

**$G_{\alpha}$ engineering and balancing of baseline *S. cerevisiae* strains.** pDAM47 and pDAM49 were amplified with oligos DAM209 + DAM212 to give the fragments $P_{RNR2}$-$GPA1$-$T_{CYC1}$ and $P_{RNR2}$-$GPA1/G_{\alpha i2}$-$T_{CYC1}$, pDAM50 and pDAM52 were amplified with oligos DAM210 + DAM212 to give the fragments $P_{ALD6}$-$GPA1$-$T_{CYC1}$ and $P_{ALD6}$-$GPA1/G_{\alpha i2}$-$T_{CYC1}$, and pDAM30 and pDAM32 were amplified with oligos DAM211 + DAM212 to give the fragments $P_{PGK1}$-$GPA1$-$T_{CYC1}$ and $P_{PGK1}$-$GPA1/G_{\alpha i2}$-$T_{CYC1}$, respectively. CPK16 was transformed with each of these fragments individually and with the gRNA plasmid ID6911 for genome integration to make strains CPK125-130. In the same way, SBY16 was engineered

with the integration of $P_{RNR2}$-$GPA1$-$T_{CYC1}$ to make strain SBY108, CPK88 was transformed with $P_{PGK1}$-$GPA1$-$T_{CYC1}$ and $P_{PGK1}$-$GPA1/G_{\alpha i2}$-$T_{CYC1}$ for CPK88, resulting in strains CPK109 and CPK112, respectively, and SBY4 with $P_{PGK1}$-$GPA1$-$T_{CYC1}$ leading to strain SBY104. CPK125-130 and CPK109, CPK112, SBY104, and SBY108 were genetically deleted for native *GPA1* by co-transformation of gRNA plasmid pDAM82 and gDNA amplicons made with oligos DAM85 + DAM235 and DAM34 + DAM236, yielding strains CPK131-136, CPK115, CPK118, SBY116, and SBY120, respectively. CPK1, CPK115 and CPK118 were transformed with the super-folding GFP ($P_{TEF1}$-$sfGFP$-$T_{CYC1}$) integration cassette purified from *Not*I-digested pEDJ26[45], and the gRNA plasmid pCfB3050, resulting in strains CPK46, CPK121 and CPK124. SBY116 was transformed with $P_{TEF1}$-$mKate2$-$T_{CYC1}$ from *Not*I-digested pDAM122, and the gRNA plasmid pCfB3048, resulting in strain SBY128. To generate SBY53, *XII-2* upstream homology, $P_{TEF1}$, *mRuby2*, $T_{ADH1t}$, and *XII-2* downstream homology were amplified with DAM39-48, respectively, from gDNA and gDAM16, and co-transformed with pCfB3048 in SBY1 for genome integration. To edit the five last amino acids in Gpa1, CPK131, or SBY120 were transformed with the gRNA plasmid pDAM6, targeting the *GPA1* C-terminal, and two overlapping homology fragments together comprising the new *GPA1/G_{\alpha}* sequence. Homology fragments were amplified from pDAM30 with the oligo combinations listed below to make the resulting strains show in parentheses:

> *GPA1*/$G_{\alpha(LCGLI)}$: DAM31 + DAM419 and DAM418 + DAM212 (CPK343),
> *GPA1*/$G_{\alpha(DSGIL)}$: DAM31 + DAM423 and DAM422 + DAM212 (CPK347),
> *GPA1*/$G_{\alpha(ETGFL)}$: DAM31 + DAM427 and DAM426 + DAM212 (CPK350),
> *GPA1*/$G_{\alpha(MCGLI)}$: DAM31 + DAM551 and DAM550 + DAM212 (CPK424),
> *GPA1*/$G_{\alpha s(QYELL)}$: DAM31 + DAM37 and DAM38 + DAM212 (SBY123).

DAM35 + ID904 amplified fragments at ~900 bp from the genomes of the resulting strains that contain *GPA1/G_{\alpha}*, which were sent for Sanger sequencing with DAM31 for sequence verification.

**Mating-type switch in *S. cerevisiae* with CRISPR/Cas9.** SBY53 and SBY128 were mating-type switched from *MAT**a*** to *MAT*α with gRNA plasmid pDAM135 to give SBY55 and SBY172. The mating-type switch was confirmed by PCR on ultrapure gDNA (Yeast DNA Extraction Kit 78870, Thermo Fisher Scientific) with oligos PR_26_D7 + PR_26_D8. Band sizes of 1.0 kb and 1.2 kb indicated *MAT**a*** and *MAT*α, respectively.

**Genome integration of heterologous GPCRs in baseline *S. cerevisiae* strains.** hGPCR expression cassettes were gel purified from respective plasmids following *Not*I digest and prior to transformation with gRNA plasmid pCfB3044. The purified pDAM71, pDAM72, and pDAM74 cassettes were each integrated with CPK131-133 to give CPK153-161, respectively, and in CPK121 to give CPK139, CPK142 and CPK331, respectively. The purified pDAM76 cassette was integrated into CPK124 and CPK134-136 to give CPK152 and CPK165-167, respectively. Purified cassettes from pDAM219-223 were each integrated into CPK131 to yield CPK450-454, respectively. The pDAM217 cassette was integrated into CPK134 to make CPK455, and purified cassettes from pDAM221 and pDAM223 were additionally integrated into CPK424 and CPK350, respectively, to make CPK456 and CPK459, respectively. The pDAM222 cassette was additionally integrated into CPK343 and CPK347 to make CPK457 and CPK458, respectively. Purified cassettes from pDAM218 and pDAM216 were each integrated into SBY123 and resulted in SBY143 and SBY146, respectively. Purified cassettes from pDAM72, pDAM71, and pDAM74 were integrated into SBY172 to give SBY156, SBY157, and SBY173, respectively.

**Construction of *S. cerevisiae* ligand production strains—serotonin, melatonin, and P-factor.** Melatonin and serotonin-producing strains were generated by genome integration of biosynthetic pathways essentially as previously described[43]. Specifically, serotonin-producing strains were created by one-pot integration of *Not*I-digested pDAM23, pDAM24, and pCfB9221, using gRNA plasmid pDAM22. The

integrations were done in SBY55 and SBY173 to give SBY91 and SBY174, respectively. The melatonin biosynthetic pathway extends directly from the serotonin pathway. Melatonin producing strain SBY92 was created from SBY91 by additional integration of *Not*I-digested pDAM25 (T$_{ADHI}$-*HsASMT* < -P$_{TEF1}$-P$_{PGK1}$ > *BtAANAT*-T$_{CYC1}$), using gRNA plasmid pCfB3020. Additionally, both serotonin and melatonin production was enhanced by multicopy random integration of Ty2-LoxP-*KlURA3*-TAG-P$_{PGK1}$-*SmTPH*-T$_{CYC1}$ in Ty2 retrotransposon sites by the transformation of *Not*I-digested pCfB2772. Screening of colonies by supernatant biosensing assays was conducted to identify the highest producers, as described in "Supernatant Assays–Quantification of Synthetic Pheromones Production Levels". This engineering was done in strains SBY91, SBY92, and SBY174 resulting in SBY138, SBY139, and SBY175, respectively. P-factor-producing strains were created by the integration of *Not*I-digested pDAM123 or pDAM236-238 using gRNA plasmid pCfB3020 into CPK152 to make CPK508-511. *Not*I-digested pDAM123 was integrated in the same way into SBY55 to make SBY155.

**Strain engineering of *S. boulardii*: construction of *ura3/his3* auxotrophies.** Transformation, incubations, and recovery after transformation were done at 37 °C as previously described[36]. *S. boulardii* ATCC MYA-796 (*Sb*.MYA-796) was co-transformed with 1 μg of gRNA_*URA3* + Cas9 plasmid pDD111 and 1.5 μg of repair template DDgb009 amplified with oligos DDpr273 and DDpr274. Transformants recovered in YPD and were plated on YPD + Nat (100 μg/ml). The resulting colonies were screened with colony PCR using oligos DDpr254 + DDpr255. The band size for successful *URA3* deletion was 430 bp and resulted in strain DD277. DD277 was then co-transformed with 1 μg of gRNA_*HIS3* + Cas9 plasmid pDD115 and 1.5 μg of repair template DDgb011 amplified with oligos DDpr276 and DDpr277. Transformants were recovered in YPD and plated on YPD + Nat (100 μg/ml). After 4 days, colonies were screened with colony PCR using oligos DDpr097 + DDpr098. The band size for successful *HIS3* deletion was 680 bp. The resulting strain DD313 was additionally verified for no growth on SC-UH dropout plates.

**Genetic deletions and G$_α$-balancing of *S. boulardii*.** All genetic modifications were performed exactly as described for *S. cerevisiae* in sections "Baseline *S. cerevisiae* strains" and "G$_α$-balancing and engineering of baseline *S. cerevisiae* strains". Strain DD313 was transformed with plasmid pDAM215 to express Cas9 (SB4) and was then genetically deleted for *SST2* (SB5) and *STE3* (SB6). SB6 and SB4 were engineered to express P$_{PGK1}$-*GPA1*-T$_{CYC1}$ (SB19 and SB21, respectively) or P$_{RNR2}$-*GPA1*-T$_{CYC1}$ (SB20 and SB22, respectively), which were all genetically devoid of native *GPA1*. *STE2* was genetically deleted in strains SB19 and SB21, which resulted in SB23 and SB25, respectively.

**Homozygous mating-type switch in *S. boulardii* with CRISPR/Cas9.** Homozygous *MAT*a/a mating-type switches were verified with oligos PR_26_D7 + PR_26_D8 as described for *S. cerevisiae* above. SB4, SB5, SB19-22, SB23, and SB25 were each transformed with 1 μg of gRNA plasmid pDAM182 to make strains SB8, SB9, SB30, SB24, SB35, SB26, SB31, and SB33, respectively.

**Reporter and hGPCR expression plasmids transformations.** One microgram of plasmid pDAM194 with the pheromone-responsive P$_{FUS1}$-*yEGFP*-T$_{CYC1}$ reporter construct was transformed in strains DD313, SB8, SB9, SB30, SB24, SB35, SB26, SB31, and SB33 to make strains SB14, SB17, and SB36-42, respectively. Next, 2 μg of hGPCR expression plasmids pDAM71, pDAM72, or pDAM74 were transformed in SB41 to make SB45-47, respectively, and in SB42 to make SB48-50, respectively.

## Experimental procedures
### Handling of synthetic pheromones and ligands.
All pheromone and ligand solutions used for dose-response and transcriptome analyses

were made as 10X concentrated stocks and never exceeded 1% DMSO content in culture (1X). Adenosine (≥99%, Sigma-Aldrich), melatonin (≥98%, Sigma-Aldrich), and serotonin hydrochloride (≥98%, Sigma-Aldrich) were dissolved in DMSO (>99.9%, Sigma-Aldrich) to a concentration of 100 mM, then tenfold diluted in SC media to 10,000 μM (10% DMSO). P-factor (TYADFLRAYQSWNTFVNPDRPNL) and α-factor (WHWLQLKPGQPMY) (Custom Peptide Synthesis, 4 mg, ≥95% purity, GenScript Biotech) were dissolved in DMSO to a concentration of 10 mM, then tenfold diluted in SC media to 1000 μM (10% DMSO). Then a tenfold dilution series using SC media + 10% DMSO was done to obtain a concentration range of 0.1–1,000 μM for all pheromones and ligands. All solutions were stored at −20 °C. For mating trials, serotonin hydrochloride, melatonin, adenosine, and combinations hereof, were dissolved in SC medium in concentrations of 1000 μM without DMSO. A tenfold dilution series was then made to obtain a concentration range of 1–1000 μM. P-factor was dissolved in DMSO and diluted in SC media, whereafter a tenfold dilution series was done in SC media + 1% DMSO to provide a concentration range of 0.1–10 μM P-factor.

**Bright-field microscopy.** Bright-field microscopy was conducted on a Leica DM4000 B microscope (Leica Microsystems) equipped with a DFC300 FX camera (Leica Microsystems).

**Dose-response and morphological analyses in *S. cerevisiae* and *S. boulardii*.** Characterization of hGPCR-signaling was done by conducting dose-response analyses. *S. cerevisiae* and *S. boulardii* strains, incubated at 30 and 37 °C, respectively, were inoculated in 0.5 ml SC or SC-UH media, respectively, for initial growth (24 h, 250 RPM). Overnight strains were diluted tenfold by the addition of 4.5 ml fresh media for further growth (20 h, 250 RPM). Prior to ligand stimulation, all cultures were adjusted to OD$_{600}$ = 0.2 in fresh media, using a P300 NanoPhotometer® (Implen), and allowed brief growth (2 h, 250 RPM). To set up ligand stimulation in 96-deep-well plates, 180 μl culture was mixed with 20 μl of each of the 10X ligand stocks resulting in final concentrations of 0–100 μM of the cognate ligands in 1% DMSO. Cultures when then incubated for sensing of ligands (4 h, 250 RPM). A signal output from P$_{FUS1}$-GFP was quantified with flow cytometry. About 30 μl of each culture was diluted in 120 μl 1X phosphate-buffered saline (PBS, Life Technologies) before sampling. *S. cerevisiae* strains were sampled on a BD LSRFortessa™ X-20 (BD Biosciences) flow cytometer, and *S. boulardii* strains on a NovoCyte Quanteon™ (Agilent) flow cytometer. For each condition, three biological replicates were analyzed, with a threshold of 10,000 events per replicate.

**Supernatant assays—quantification of synthetic pheromones production levels.** Production strains were inoculated in 0.5 ml SC media for initial growth (24 h, 30 °C, 250 RPM). Production cultures were then set up by replacing culture media by pelleting (2500 × *g*, 3 min) and resuspension in 500 μl fresh SC. Cultures were then incubated for production in 96-deep-well plates, which for HPLC was done with an initial OD$_{600}$ = 0.2 (24 h, 30 °C, 250 RPM) and for biosensing an initial OD$_{600}$ = 2.0 (5 h, 30 °C, 250 RPM). To acquire the pure media supernatant for quantification of ligand production, cells were removed by pelleting (5000 × *g*, 5 min) and extraction of supernatant by pipetting twice. HPLC analysis of serotonin and melatonin amounts produced by yeast strains was done on the Thermo Scientific™ UltiMate™ 3000 HPLC using the Agilent Zorbax C18 4.6 × 100 mm 30l5-Micron column with a Phenomenex AFO-8497 filter. Solvent A was 0.05% acetic acid, solvent B Acetonitrile. Data analysis was done using Chromeleon™ Chromatography Data System (CDS) Software. Serotonin and melatonin values of samples were determined according to standard curves of serotonin hydrochloride and melatonin, in the range of 10 to 150 uM and 2 to 20 uM, respectively. Serotonin peaks were detected at 1.920 min, while melatonin peaks were seen at 7.037 min.

## Mating trials

Mating trials were based on the fluorescent detection of mated and haploid cells by flow cytometry. This was accomplished by always partnering SBY-strains expressing mRuby2 or mKate2 with CPK-strains expressing sfGFP, providing a double-fluorescent signal from diploid cells. Firstly, strains were individually inoculated in 0.5 ml SC for initial growth (24 h, 30 °C, 250 RPM), then strains were diluted tenfold for further growth (22 h, 30 °C, 250 RPM). To remove accumulated pheromones of the pre-mating cultures, media was removed by pelleting (5000×$g$, 4 min.) and resuspended in fresh SC media without DMSO. To set up the mating trials, the cell densities of individual cultures were first determined by flow cytometry: 10 µl culture was diluted in 190 µl 1X PBS and run on a NovoCyte Quanteon and volumes containing 1,000,000 cells were then calculated. The individual strains were then mixed to form co-cultures at an initial 1:1 ratio of 1,000,000 cells of each of the two mating strains into a volume of 200 µl SC media per replicate (i.e., a total of 10,000 cells/µl) in 96-deep-well plates. At this step, media containing synthetic pheromone or ligand was supplemented during resuspension. All mating trials had a 5 h incubation time at 30 °C and 250 RPM. After mating, the co-cultures were examined by flow cytometry with 30 µl culture diluted in 120 µl 1X PBS sampled on a NovoCyte Quanteon™ (Agilent) flow cytometer, and diploid selective plating was done on SC-UW. All mating trials were done in five replicates, with a threshold of 50,000 events per replicate.

## Data analysis and statistical analysis

All analyses are detailed in Supplementary Data 9–14.

**Flow cytometry data and gating.** All flow cytometry data were extracted as FCS files and gated in FlowLogic™ v8.3 (Inivai Technologies). SSC-A, FSC-A, and fluorescence data points were derived from median values and median fluorescence intensities (MFI) of gated populations. Normalized MFI (nMFI) was obtained by normalizing it to the mean of background MFIs. For statistics and data analysis, means of medians and MFIs were applied ± standard deviation. Population proportions were derived directly from event counts per gate, including diploid frequencies in mating trials. For $P_{FUS1}$-GFP-based hGPCR characterization and biosensor assays, small non-responsive cells were removed by consistent exclusive gates based on minimum FSC-A values (Supplementary Fig. 6A). In the case of normal-sized, but non-responsive, cells, their distinct autofluorescence was used to create an exclusive gate (B-530/30 vs Y-615/20) (Supplementary Fig. 6B). For mating trials, firstly, compensation was done between Y-615/20 and B-530/30 using CompLogic automatic compensation in FlowLogic™ v8.3 (Inivai Technologies). Secondly, singlets were gated based on SSC (SSC-A vs SSC-H) (Supplementary Fig. 6C). Lastly, a quadrant gate was applied in the double-fluorescence dimension (Y-615/20 vs B-530/30) to classify all detected cells as either identifiable haploid A, haploid B, or diploid, or alternatively unidentifiable unknown cells lying below fluorescence thresholds. The quadrant gate was established by employing fluorescence minus one (FMO) control (Supplementary Fig. 6C).

**Mating efficiencies and diploid frequencies.** Diploid frequencies are the proportion of the entire sampled mating culture that are identified as diploids. Mating efficiencies describe how efficiently the given number of diploids have formed relative to the amount of potential mating events. In this study, mating efficiency is determined by the limiting haploid's method, which assumes that each diploid must have originated from exactly one cell of each of the two mated haploids. With this assumption, the number of diploids that can be formed is constrained by the haploid with the fewest available cells for mating— i.e., the limiting haploid. Applied to flow cytometric mating assays

using double-fluorescence, this translates to one singlet double-fluorescent diploid must have originated from one singlet haploid A of fluorescence A (e.g., GFP) and one singlet haploid B of fluorescence B (e.g., mRuby2). The mating efficiency is described by the population frequencies of diploids and haploids as given by Eq. (1), here exemplified by haploid A being limiting.

$$\text{If haploid A is limiting, then}:$$
$$\text{Mating Efficiency} = (\text{Diploids})/(\text{Haploid A} + \text{Diploids})*100 \quad (1)$$

**Statistical and data analysis.** Data analysis, statistical analysis, and graphing was done in RStudio v1.4.1106 for R (v4.1.2), GraphPad Prism v9.2.0 (GraphPad Software), and FlowLogic™ v8.3 (Inivai Technologies). Sigmoidal dose-response curve fits were computed on nMFIs using nonlinear regression by the variable slope (four parameters) model in GraphPad Prism v9.2.0 (GraphPad Software), using default settings (Supplementary Data 9). The significance of $P_{FUS1}$-GFP reporter expression was assessed by one-way or two-way ANOVA multivariate test and post-hoc analysis by Dunnett's or Tukey's multiple comparison test, respectively (Supplementary Data 9, 10). Multivariate analysis by determination of Pearson correlation coefficients for SSC-A, FSC-A and GFP MFIs and simple linear regression was done in GraphPad Prism v9.2.0 (GraphPad Software), using default settings (Supplementary Data 11). Population SSC-A histograms were overlaid and normalized to 100% using FlowLogic™ v8.3 (Inivai Technologies). To assess the significance of mating efficiencies, a two-way ANOVA multivariate test and post-hoc analysis by Dunnett's multiple comparisons test was done (Supplementary Data 12–14). All statistical tests were done in GraphPad Prism v9.2.0 (GraphPad Software) with a default 95% confidence interval ($\alpha = 0.05$) applied and multiplicity-adjusted $p$ values were reported to account for all multiple comparisons within tests.

## Flow cytometry settings

**BD LSRFortessa™ X-20 (BD Biosciences) settings.** Excitation was done with a blue 488 nm laser and GFP emission was detected with a 530/30 nm bandpass (BP) filter (B-530/30) (524 V), while FSC was detected with a photodiode detector with a 488/10 nm BP filter (240 V) and SSC was detected with a photomultiplier tube (PMT) with a 488/10 nm BP filter (267 V).

**NovoCyte Quanteon™ (Agilent) with NovoSampler Q (Agilent) settings.** Excitation was done with a blue 488 nm laser for GFP and a yellow 561 nm laser for mRuby2, mKate2, FSC, and SSC. GFP emission was detected with a 530/30 nm BP filter (471 V), mRuby2 and mKate2 was detected with a 615/20 nm BP filter (616 V), while FSC and SSC was detected with a 561/14 nm BP filter (400 V). The machine ran with a core diameter of 10.1 µm (24 µl/min), two mixing cycles every well (2000 g, Acc. = 1 s, Dur. = 10 s), one rinse cycle every well, and a threshold for event detection at >150,000 FSC-H. NovoFlow Solution (Agilent) was used and the machine was calibrated with NovoCyte QC Particles (Agilent).

## Procedure for transcriptome analysis of biosensing yeast strains

**Yeast cell harvest and RNA purification.** Strains CEN.PK2-1C, CPK2, CPK139, CPK152, and CPK331 were each inoculated in 0.5 ml SC medium and incubated O/N at 30 °C and 250 RPM. All cultures were then diluted tenfold by the addition of 4.5 ml SC medium 20 h prior to $OD_{600}$-adjustment and left for incubation at 30 °C and 250 RPM. Each culture was adjusted to $OD_{600} = 1.0$ the following day and left for an additional 2 h of incubation before aliquoting 180 µl into three wells each in a 96-deep-well plate. About 20 µl of SC alone, or with tenfold increasing concentrations of cognate ligands, and 10% DMSO was added to each 180 µl aliquot and left for shaking incubation at 30 °C

and 250 RPM for 4 h prior to harvest. Total RNA was purified using the RNeasy Mini Kit (QIAGEN) according to the protocol.

**Library preparation and sequencing.** Purified total RNA was quantified using a Qubit fluorometer and qualified using a Fragment analyzer. We constructed mRNA libraries using Truseq stranded mRNA kit (Illumina) and TruSeq RNA CD indexes (Illumina) according to the kit protocols. Libraries were sequenced as paired ends on an Illumina NextSeq 500 at 75 bp × 2, with NextSeq High output 150 cycles kit. All raw sequencing data were deposited into the NCBI Short Read Archive under the accession number (PRJNA790752).

**Transcriptome analysis.** All the raw sequencing reads were trimmed using Trimmomatic[46] to filter sequencing adapters and low-quality reads. The clean reads were aligned against yeast reference genome using STAR[47] with the parameters: "--outFilterMultimapNmax 100 --alignSJoverhangMin 8 --alignSJDBoverhangMin 1 --outFilterMismatchNmax 999 --alignIntronMax 5000 --alignMatesGapMax 5000 --outSAMtype BAM Unsorted". The genome reference and gene annotations of *Saccharomyces cerevisiae* (R64-1-1) were obtained from Ensembl. We used HTSeq-count[48] to calculate raw counts for each yeast gene, with the parameter "-s reverse" to specify the strand information of reads. We applied the TMM (trimmed mean of M values) method[49] implemented in edgeR package to normalize gene expressions. The TMM normalized RPKM (Reads Per Kilobase of transcript, per Million mapped reads) values were further log2 transformed to generate heatmap and other plots using pheatmap and ggplot2[50].

**DEG analysis.** Genes with cpm (counts per million mapped reads) above 1 in at least three samples were kept for the DEG analysis[51], which was performed by the "exactTest" function in edgeR between each pair of conditions[52]. We identified DEGs in each pairwise analysis using the standard: log2-transformed fold change above 1 and FDR under 0.05. We applied TopGO[53] to perform GO enrichment analysis. Benjamin-Hochberg false discovery rate correction was used to adjust *P* values to determine the significant enriched GO terms.

**Reporting summary**
Further information on research design is available in the Nature Research Reporting Summary linked to this article.

## Data availability
All raw sequencing data used for transcriptome analysis were deposited into the publicly available NCBI Short Read Archive under the accession number (PRJNA790752). There are no restrictions on data availability, and unique identifiers are listed in relevant tables. Source data are provided with this paper.

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

## Acknowledgements

This study is supported by grants from the Velux Foundations (28309) to E.D.J. and the Novo Nordisk Foundation (NNF20CC0035580 and NNF19CC0035454) to M.K.J. and B.L., the National Science Foundation (CBET-1934284), seed funding from the National Institutes of Health (P30DK034987), and Startup Funds from North Carolina State University to N.C., a T32 training grant to J.M.V. (5T32GM008776), and a fellowship from the Ministry of Higher Education—Oman to ISA. We thank Frederik B.F. Neergaard, Lara P. Munkler, Francisco J.A. Rios, and Chenxi Zhang for technical support on flow cytometry. We thank Dr. Nicholas Milne for providing plasmid pCfB9221. Schematic figure illustrations for panel A in Figs. 1, 2, and 4, and for panels A and G in Fig. 5, were created with BioRender.com.

## Author contributions

E.D.J.: Project administration, equal, Conceptualization, lead, Writing—original draft, lead, Formal analysis, equal, Writing—review & editing, equal, Funding acquisition, equal. M.D.: Investigation, lead, Conceptualization, equal, Formal analysis, equal, Writing—review & editing, equal. X.M.: Investigation, equal. R.U.V.: Investigation, supporting. G.S.: Investigation, supporting. M.B.R.: Investigation, supporting. B.L.: Investigation, supporting. J.M.V.: Investigation, supporting. D.D.: Investigation, supporting. S.P.H.: Investigation, supporting. I.Al'.A.: Investigation, supporting. J.Z.: Investigation, supporting. N.C.: Conceptualization, supporting, Funding acquisition, supporting. Writing—review & editing, supporting. M.K.J.: Project administration, equal, Conceptualization, equal, Writing—original draft, equal, Writing—review & editing, equal, Funding acquisition, equal. All authors approved the manuscript.

## Competing interests

The authors declare no competing interests.
