## [Peer Review File · Nature Communications]

Reviewers' Comments:

Reviewer #1:

Remarks to the Author:

This manuscript builds on a long history, dating back to the early 1990s, showing that human G protein coupled receptors can substitute for the pheromone receptors in yeast *S. cerevisiae*. Such heterologous expression systems may be useful for drug screening and directed evolution, among other applications. This paper confirms well-established and published data for four human receptors expressed in *Saccharomyces cerevisiae* (refs. 6,8,11,21) and show that the approach can be ported to another closely related yeast, *Saccharomyces boulardii*.

Figure 1 shows transcription reporter (Fig. 1B) and transcriptomics (Fig. 1C/D) for the four human GPCRs, and their cognate ligands, expressed in *S. cerevisiae*. Please provide (i) transcriptomics data for P-factor in panels C and D (as done in panel B), (ii) calculated EC50s for each ligand (and state how those values compare with EC50s in native/human cells), (iii) positive control data for the native yeast ligand (alpha-factor) in WT and *sst2* cells in panel B (as done in panels C and D), (iii) negative control data using either non-cognate ligand-receptor pairings or a competitive antagonist or both. These peptides and amino acid metabolites could have effects on yeast that have no relationship to GPCR binding (there are 10-53 DEGs unique to strains treated with human ligands - are they receptor dependent?). The transcriptomics data is new but the information is not used to draw any biological or pharmacological conclusions. The authors state on p. 3 that the lack of transcriptomics data has been "constraining yeast as a model organism" but I'm not clear how the data are in any way useful.

Figure 2 shows transcription reporter data for three of the four human GPCRs, and their cognate ligands, expressed in *S. boulardii*. Please plot the data in the same manner as in Figure 1B and please provide data for serotonin. Again I don't really understand how studies in two yeasts are in any way better than studies in one. Please explain how "this could have major implications to advance the engineering of yeast therapeutics" as claimed.

Figure 3 shows cell morphology data for the four human GPCRs, and their cognate ligands, expressed in *S. cerevisiae*. Please provide positive control data for the native ligand alpha-factor in WT and *sst2D* cells and negative control data using either non-cognate ligand-receptor pairings or a competitive antagonist or both. Panel D should show more cells. The morphology data correlate with the transcription data, but the transcription data looks much cleaner. This adds little to the story.

Figure 4 shows cell mating data for the four human GPCRs, and their cognate ligands, expressed in *S. cerevisiae*. Please provide positive and negative control data.

Figure 5 shows cell mating data for cells expressing human GPCRs and engineered to secrete their cognate ligands. Panel B data are convincing. Panels C, D and E are not convincing and should be removed. Panel E would be a nice result if serotonin+P-factor was better than serotonin alone. This is potentially the most innovative part of the paper but the data are not very strong. Perhaps the experimental design could be optimized but as presented it is not very persuasive - it might help to use SST2+ cells, since deletion of this gene reduces mating efficiency.

As noted by the authors on p. 3, MAPK activation is another important readout of mating pathway activation. Do the human receptors activate MAPK?

Minor.

p. 2 Ste2 senses 'alpha' factor, not 'a' factor as written.

Legend to Figure 5 is duplicated.

The experiments are done in *sst2* deletion strains but the rationale for doing so is not clear and the normal function of SST2 is not stated.

p. 6 "expression should ideally be high in the absence of ligand and subsequently decrease upon ligand recognition to facilitate Gbeta/gamma release" – this statement makes no sense, Gbeta/gamma is released upon receptor activation regardless.

p. 7 "snipper" not "sniper"

Reviewer #2:

Remarks to the Author:

Jensen et al presented the rational design of heterogeneous GPCR-based cell-cell communication system and used it to control mating pathway activation in *S. boulardii*. They first explored the transcriptomic changes when replacing the wild-type GCPR with four heterogeneous ones, and then engineered the *S. boulardii* based on the knowledge of *S. cerevisiae*, found the morphologies were regulated by four different pheromones, eventually constructed synthetic hGPCR-based cell-cell communication system and to control yeast mating pathway. It was an interesting topic to explore the new intercellular signaling pathway in a probiotic yeast (*S. boulardii*), even though the same topic had been extensively studied in both bacterium and yeast (*Nature communications*, 2018, 9(1): 1-12; *Nature communications*, 2020, 11(1): 1-11). Before considering to be published, I have several major concerns about the manuscript.

1, In line 53-57, "investigations of the compatibility of hGPCR with yeast mating have been limited...." need to give the proper reasons of the transcriptomic investigation for artificial system. The following design of the synthetic mating pathways seems not use the transcriptomic data.

2, In line 165-167, why just estimated A2bR, MT1, and Mam2, but not involving 5-HT4b. the author need to present their reasons, since the following Figure still used the P-factor and 5HT4b system.

3, In line 195, the data of FSC-A was not shown in Figure 3A.

4, In line 296-297, Why chose Serotonin and P-factor to construct the autonomous yeast mating system in Figure 5E, but only shown the serotonin, Melatonin and Adenosine three synthetic pheromones in Figure 5A-5D, not mention the data about the P-factor. The logic of the given data is weird, and uneasy to be following.

5, The colors and the sizes of triangle, square and circle is not easily distinguished with each other.

REVIEWER COMMENTS

Reviewer #1 (Remarks to the Author):

This manuscript builds on a long history, dating back to the early 1990s, showing that human G protein coupled receptors can substitute for the pheromone receptors in yeast *S. cerevisiae*. Such heterologous expression systems may be useful for drug screening and directed evolution, among other applications. This paper confirms well-established and published data for four human receptors expressed in *Saccharomyces cerevisiae* (refs. 6,8,11,21) and show that the approach can be ported to another closely related yeast, *Saccharomyces boulardii*.

Figure 1 shows transcription reporter outputs (Fig. 1B) and transcriptomics (Fig. 1C/D) for the four human GPCRs, and their cognate ligands, expressed in *S. cerevisiae*. Please provide (i) transcriptomics data for P-factor in panels C and D (as done in panel B), (ii) calculated EC50s for each ligand (and state how those values compare with EC50s in native/human cells), (iii) positive control data for the native yeast ligand (alpha-factor) in WT and *sst2* cells in panel B (as done in panels C and D), (iv) negative control data using either non-cognate ligand-receptor pairings or a competitive antagonist or both.

(v) These peptides and amino acid metabolites could have effects on yeast that have no relationship to GPCR binding (there are 10-53 DEGs unique to strains treated with human ligands - are they receptor dependent?). The transcriptomics data is new but the information is not used to draw any biological or pharmacological conclusions. The authors state on p. 3 that the lack of transcriptomics data has been “constraining yeast as a model organism” but I’m not clear how the data are in any way useful.

>> We thank the reviewer for reading and reviewing our manuscript. For Figure 1 we here provide our responses:

Ad (i): This is a relevant comment. In our original manuscript we presented transcriptomics data for the strains expressing three out of four heterologous GPCRs in comparison to native STE2-based GPCR signalling in both WT strain and the sensitized *sst2* deletion background. The ligands used in that original 60-samples RNAseq study included both small-molecule chemistries and peptides as agonists as well as all human GPCRs, and documented to the best of our knowledge for the first time the common core signalling-responsive genes differentially regulated upon human GPCR expression.

In our revised manuscript we now present new data on GPCR specificities, as well as positive and negative control experiments provided for all four studied heterologous GPCRs (as also requested by this reviewer, see below). Following reading and reviewing of our revised manuscript we hope the reviewer will agree with us that performing another 12-sampled RNAseq experiment only for the Mam2-expressing strain will be technically challenging to compare with the existing omics data set. Furthermore, and especially in relation to the new data presented on specificities and controls for all four GPCRs (see data for new Suppl. Fig. 1B, Suppl. Fig 1D, and Fig. 5), we hope the reviewer will agree with us that more transcriptomics is an excessive undertaking for the mere reason of consistency/systematics.

Ad (ii): Very relevant point. We have now included calculations of EC50s for all human GPCRs expressed in yeast and compared those to the EC50s reported from reporter/activity assays carried out in human cells. The EC50s obtained in yeast are included in the new Suppl. Table S1 .

We have added the following sentence in the revised manuscript as well:

“Importantly, and corroborating previous work on fungal GPCRs (Billerbeck et al. 2018), hGPCRs are orthogonal across their non-cognate ligands when expressed in yeast (Suppl. Fig. S1D), while the EC50s observed for MT1 and 5-HT4b are approximately two orders of magnitude higher in yeast cells, and approximately one order of magnitude lower for A2bR in yeast cells compared to mammalian cells (CHO)(Suppl. Table S1)”

Ad (iii): Ok. In line with other studies having characterized Ste2-responsiveness to alpha-factor in WT and *sst2* cells (e.g. Shaw *et al.*, 2019) we now also include our data on Ste2 in our revised Suppl. Fig. S1B.

Ad (iv): This is a relevant point, and indeed ligand-promiscuity and biased signalling is prevalent among human GPCRs. In our revised manuscript we now provide new data with full cross-comparison of all 5 ligands against all 5 GPCRs studied. This also includes negative controls. Please find the new data in Suppl. Fig S1D highlighting strong ligand specificity among the tested GPCRs characterised in this study. Main text and legend for Suppl. Fig. S1 has now been updated as follows:

From

“B. Fold change from 0-1000 μ M serotonin supplementation in strain CPK165. C. Coupling-shifts are presented as log-scaled fold changes in fluorescence from a PFUS1-GFP reporter following hGPCR integration (+hGPCR: strains SBY143, SBY146, CPK153, CPK156, CPK159, CPK165, and CPK450-459) over background (no hGPCR: SBY123, CPK131, CPK134, CPK343, CPK347, CPK350, and CPK424). Specific $G\alpha$ subunits are indicated for each hGPCR presented in the plot. Means and standard deviations represent three biological replicates. D.”

to

*“(B). Native Ste2-signaling in wild-type (CPK86) and sensitized *sst2* Δ (CPK503) yeast strains from increasing concentrations of supplemented pheromone (α -factor). The FUS1 promoter was used to drive expression of a GFP reporter in response to supplemented pheromone. AU: artificial units. (C). Fold-change from 0-1000 μ M serotonin supplementation in strain CPK165. (D). A 6 x 6 orthogonality matrix was made by supplementation of 10 μ M of each indicated ligand to individual heterologous GPCRs contained in strains CPK155, CPK158, CPK161, CPK167 and wild-type Ste2 (CPK86). GPCR-signaling was determined from mean fluorescence intensities (MFI) with a GFP*

reporter. No GPCR (CPK133) and no ligand controls were included as shown. Data represent means of three biological replicates for all samples. (E).”

Ad (v): On whether the 10-53 DEGs are unique to expression of the three GPCRs or whether they are ligand-specific, we now present a further analysis on our transcriptomics data set. Here we show the fraction of GPCR-specific DEGs (any heterologous GPCR vs native STE2) without ligand addition (0 μ M), and compare this to the fraction of GPCR-specific DEGs (any heterologous GPCR vs native STE2) in the presence of ligands addition (1-100 μ M). From this we find the following:

Statistics

	Specific DEGs	DEGs in 0 μ M	DEGs in ligands
MT1	10	0	1
5HT4	53	21	35
A2bR	13	1	8

This means that for 5HT4-specific DEGs almost half of the DEGs are responding to expression of 5HT4, whereas for A2bR and MT1, one and zero DEGs, are identified by mere expression of the receptors. In short most DEGs arise from ligand addition in hGPCR-expressing strains. We have now inserted a sentence on this in the manuscript.

Also, importantly we show that there is no differential expression of the *FUS1* promoter upon supplementation of any of the ligands used in this study (please see new matrix plot in Suppl. Fig. S1D).

Figure 2 shows transcription reporter data for three of the four human GPCRs, and their cognate ligands, expressed in *S. boulardii*. Please plot the data in the same manner as in Figure 1B and please provide data for serotonin. Again I don't really understand how studies in two yeasts are in any way better than studies in one. Please explain how “this could have major implications to advance the engineering of yeast therapeutics” as claimed.

>> Good point. We have now plotted the data in the same line chart manner as in Figure 1. Importantly, we want to highlight that this data is the first demonstration of porting in human GPCRs in a mating-activated *S. boulardii*. On the reasoning for the importance of demonstrating mating pathway activation in diploid probiotic yeasts, we would like to clarify the importance of our finding. *S. boulardii* is a marketed probiotic used to treat gastrointestinal disorders (e.g. Kelesidis & Pothoulakis, 2012, PMID: 22423260). Yet, as *S. boulardii* is a native diploid it has never been shown if/how the mating pathway of this conspecific yeast could be activated. Without this, GPCR signalling is hard to engineer and apply in yeast as theranostic tools, as otherwise recently demonstrated for *S. cerevisiae* (Scott et al., *Nature Medicine*, 2021, PMID: 34183837). Thus, as *S. cerevisiae* does not colonise

intestinal guts of mammals, enabling GPCR-based sensing and “payload” delivery via an activated mating pathway could hold immense opportunities for furthering the application of *S. boulardii* as an advanced probiotic. We hope this background information clarifies why “two yeasts are better than one”. In the revised manuscript we have updated the main text related to the motivation of the experiment presented in Figure 2.

We hope the reviewer agrees with us that the demonstration documented with human A2bR, MT1, and fungal Mam2 sufficiently justifies our rationale and conclusions. In the rationale for our study we have now included the following text, which we hope addresses the reviewer’s uncertainty related to our wish to showcase that mating pathway activation in two yeasts rather than one:

*“However, in the diploid *S. boulardii*, hGPCR-signaling has never been reported, even though this could have major implications to advance the engineering of yeast therapeutics 13, especially as novel theranostics enabling both GPCR-based diagnosis as well as in situ therapeutics delivery based on the active mating pathway.”*

Figure 3 shows cell morphology data for the four human GPCRs, and their cognate ligands, expressed in *S. cerevisiae*. Please provide positive control data for the native ligand alpha-factor in WT and *sst2D* cells and negative control data using either non-cognate ligand-receptor pairings or a competitive antagonist or both. Panel D should show more cells. The morphology data correlate with the transcription data, but the transcription data looks much cleaner. This adds little to the story.

>> Valid point. As mentioned above, we now have provided positive control data for WT and *sst2* mutant strains in response to alpha-factor, as well as new data on the specificities of mating pathway activation in yeast upon expression of single heterologous GPCRs (see new Suppl. Fig. S1B).

Regarding the morphology, we agree that the abundance of transcripts for the core DEGs shown in Fig. 1C-D (and now also used to optimise full synthetic mating - new Fig. 5E) correlates with ligand dosage. However, before deleting Figure 3D we wish to remind the reviewer that the main reason for presenting representative data on morphology on strains designed for different expression levels of GPCRs, is to complement the finding from the RNAseq data, focusing on different ligand levels. Together, promoter strength and ligand dosage, this information is used for us to make decisions about optimal design and condition for the synthetic mating trials in Fig. 4 and Fig. 5. Thus, we hope the reviewer agrees with us that Fig. 3D belongs in the manuscript.

Figure 4 shows cell mating data for the four human GPCRs, and their cognate ligands, expressed in *S. cerevisiae*. Please provide positive and negative control data.

>> We are sorry about the confusion. We realise we have not been clear enough in our original Figure 4 legend. In our revised manuscript we have revised the legend to clearly stipulate what the negative and positive controls are referring to. Indeed, both positive and negative data sets for all four tested GPCRs and their cognate ligands are presented. As for

positive controls (light green bars; CPK46 s. SBY55), we spike in the single ligands in the cultivation medium, while for negative controls (light pink bars), we used strains w/o heterologous GPCRs (CPK121 or CPK124 vs. SBY55). Furthermore, for negative controls, we also include data for all synthetic mating trials w/o supplementation of ligands (indicated by (-) in the x-axis). We hope the updated Figure 4 legend clarifies to the reviewer that positive and negative control data indeed is already presented. The figure legend was updated as follows:

From:

“Schematic outline showing production, supplementation, and sensing of melatonin (MT₁), serotonin (5-HT_{4b}), or P-factor (Mam2) in yeast mating pairs. Adenosine (A2bR) was present solely from supplementation” to “Schematic outline showing production, supplementation, and biosensing of melatonin (MT₁), serotonin (5-HT_{4b}), or P-factor (Mam2) in yeast mating pairs. Adenosine (A2bR) was introduced solely by supplementation as illustrated”, and from “CPK46xSBY55 and CPK121xSBY55 or CPK124xSBY55 were used as references for positive and negative mating pair controls with relevant ligand supplementation, respectively.”

to

“CPK46xSBY55 were used as positive control for native mating throughout. CPK121 (GPA1) and CPK124 (GPA1/G_{ai2}) do not contain any GPCRs and were used as negative mating controls with SBY55 in panel B-D (CPK121) and in panel E (CPK124).”

Figure 5 shows cell mating data for cells expressing human GPCRs and engineered to secrete their cognate ligands. Panel B data are convincing. Panels C, D and E are not convincing and should be removed. Panel E would be a nice result if serotonin+P-factor was better than serotonin alone. This is potentially the most innovative part of the paper but the data are not very strong. Perhaps the experimental design could be optimized but as presented it is not very persuasive - it might help to use SST2+ cells, since deletion of this gene reduces mating efficiency.

>> We thank the reviewer for considering the Figure 5 data innovative. This data indeed represents the goal of the entire study. We also agree that the data presented in the final “exam” of full synthetic mating left room for improvement. Thus, as also suggested by the reviewer, we undertook the study of several experimental parameters to try to optimize the mating efficiency.

First, considering that native yeast mating relies on pheromone gradient sensing, we assessed if changes in the ratio of total numbers of cells (as measured by cell counting) from each of the two mating types would provide improved pheromone gradients and thus optimize the chemotactic behaviour leading to mating. Indeed, for all the synthetic mating pairs of the original Figure 5C-E, we observe that changing the ratio of cells for each of the two mating types can improve the average mating efficiency ~3-fold from ~8.5% to ~28% (serotonin and

adenosine; new Fig. 5E) and ~2-fold from ~9% to ~17% (serotonin and melatonin; new Fig. 5F).

Next, as pheromone production in native reference budding yeast is regulated by inducible promoters (e.g. *MFA1*), we sought to exploit the RNAseq data generated in this study to search for promoters induced following GPCR activation. Indeed, here we found promoters *MFA1*. While *MFA1* is among the core DEGs identified in Fig 1C-D, it shows a relative high background expression in the absence of mating pathway activation (Fig. 1D). To further increase the absolute range of expression of our synthetic pheromones, we therefore decided to include both *FUS1* and *AGA2* promoters showing low background expression and relative high induction upon mating pathway activation (Fig. 1D). In comparison to these we also re-did the mating trials with a constitutive *TEF1* promoter driving the expression of the synthetic pheromone P-factor. Combining the parameters of ratio tuning and inducing pheromone expression upon mating pathway activation improved autonomous mating by ~3.5-fold to a maximum mating efficiency of 26%. We hope the reviewer agrees with us that obtaining the high percentages of full synthetic and autonomous mating are a significant improvement of the previous results.

All new synthetic mating data has been included in a revised Figure 5.

Legend for Figure 5 was updated:

from “**E.** Paracrine signaling from serotonin and P-factor production in an autonomous yeast mating pair. Autonomous mating between strains sending and receiving P-factor and serotonin: CPK152xSBY173 (no production), CPK152xSBY175 (production of serotonin only), CPK448xSBY173 (production of P-factor only), CPK448xSBY175 (production of both serotonin and P-factor). CPK124xSBY172 (No GPCR) without hGPCRs (Mam2 and 5-HT_{4b}) was included as negative control. Means and standard deviations represent five biological replicates. Statistical significance was determined for mating efficiencies using one-way (B-D) and two-way (E) analysis of variance (ANOVA) with Dunnett’s multiple comparisons test, relative to the negative control in each setup, using GraphPad Prism (**p ≤ 0.01) (see Methods).”

to

“**E-F.** Strain ratios of 10:1, 1:1, and 1:10 were investigated for mating pairs CPK152xSBY157 (E) and CPK152xSBY156 (F) in 5 hrs co-incubations with 0 μM, 10 μM, or 100 μM adenosine and serotonin (E) or melatonin and serotonin (F). **G.** Paracrine signaling from biosensing of serotonin and P-factor production in an autonomous yeast mating pairs. SBY175, expressing the GPCR Mam2 and producing serotonin, was crossed in ratios 1:10, 1:1, and 10:1 with strains expressing the GPCR 5-HT_{4b} and producing P-factor constitutively from *P_{TEF1}* (CPK508), or in dynamically regulated response to serotonin from the pheromone-inducible promoters *P_{AGA2}*, *P_{MFA1}*, or *P_{FUS1}* (CPK509-511). CPK124xSBY172 with no GPCRs or production were used as negative controls. Bars represent averages from three biological replicates. Statistical significance was determined for mating efficiencies using one-way (**B-F**) and two-way (**G**) analysis of variance (ANOVA) with Dunnett’s multiple

*comparisons test, relative to the negative control in each setup, using GraphPad Prism (** $p \leq 0.01$) (see Methods)."*

We inserted main text for the new Figures 5E+F by inserting:

"Although mating efficiency for strains that sense melatonin and adenosine reached 40% (Fig. 5B), mating efficiency only reached ~10% at best for strains sensing serotonin and adenosine or serotonin and melatonin (Fig. 5C & 5D). As noticed earlier, biosensing strains elicit different responses depending on the hGPCR they express (Fig. 1B, Fig. 3, Fig. 4), and it therefore appeared likely that tuning of the ratio would improve the mating efficiency. During supplementation of both ligands (10 μ M & 100 μ M) we discovered that 10x ratio tuning improved mating efficiency by ~3-fold for the serotonin and adenosine mating pair (Fig. 5E), and ~2-fold for the serotonin and melatonin mating pair (Fig. 5F). In this way, ratio tuning can be used to increase mating efficiency by several fold for full synthetic mating."

And for Fig. 5G (formerly Fig. 5E) we replaced

"Here we observed the highest mating efficiency (6.6%) in the mating pair producing both synthetic pheromones. Also, mating efficiency increased compared to the negative mating control when at least one of the synthetic pheromones was produced (Fig. 5E). Once again, resulting diploids showed altered SSC-A following hGPCR stimulation (Suppl. Fig. S4B & S4C). Lastly, and worth noting, is that while the mating efficiency was lower for mating pairs in which only one synthetic pheromone was produced, compared to the efficiency observed for mating pairs producing both serotonin and P-factor, the diploid frequency was highest when only serotonin was produced (Fig. 5E)."

with

"Here, serotonin was produced continuously from one strain (SBY175) while P-factor was expressed in a partner strain (CPK508-511), either constitutively from P_{TEF1} or dynamically regulated from one of the pheromone-responsive promoters P_{AGA2} , P_{MFA1} , or P_{FUS1} (Fig. 1D, Suppl. Table S2) during 5-HT_{4b} signaling. Once again, we applied ratio tuning to increase mating efficiency and found that a 10x surplus of SBY175 relative to each CPK strain gave the highest mating efficiency in every trial (P_{TEF1} : 9%; P_{AGA2} : 12%; P_{MFA1} : 15%; P_{FUS1} : 26%) (Fig. 5G). Thus, the highest mating efficiency resulted from engineering of dynamically regulated P-factor expression from P_{FUS1} which, in combination with ratio tuning, gave ~3.5-fold improvement over the negative control lacking both GPCRs and paracrine signaling."

Furthermore, Discussion was updated with:

"Most importantly, simple ratio tuning between synthetic mating pairs enabled 2- and 3-fold mating efficiency improvements over 1:1 ratios at supplementation with both ligands

(Fig. 5E & 5F), and up to 3.5-fold mating efficiency improvement for autonomous mating pairs when serotonin was endogenously produced and dynamically regulated pheromone-inducible P-factor expression was introduced(Fig. 5G)."

As noted by the authors on p. 3, MAPK activation is another important readout of mating pathway activation. Do the human receptors activate MAPK?

>> Clarification needed here. The yeast mating pathway indeed employs a canonical MAPK cascade, including MAPK kinase kinase (Ste11) which phosphorylates and activates a dual specificity MAPK kinase (Ste7), and the MAP kinase (Fus3). In native budding yeast, upon pheromone-induced activation of the mating pathway, the MAPKs are indeed activated via phosphorylation (e.g. Hurst & Dohlman, JBC, 2013, Fig 5a, PMID: 23645675). Likewise, for reporter gene activation (e.g. most often used *FIG1* or *FUS1* promoters) to occur upon binding of ligands to either native GPCRs or heterologous ones, the MAP kinase FUS3 must be phosphorylated (Blackwell et al., 2007, BMC Cell Biol., PMID: 17963515) to translocate to the nucleus and initiate reporter gene expression via the transcription factor STE12. So in short, prior art strongly suggests that activation of human receptors mediate MAP kinase phosphorylation.

Minor.

p. 2 Ste2 senses 'alpha' factor, not 'a' factor as written.

>> Correct. Thanks for noting this.

Legend to Figure 5 is duplicated.

>> Thanks. Duplicate has now been deleted.

The experiments are done in *sst2* deletion strains but the rationale for doing so is not clear and the normal function of SST2 is not stated.

>> Good point. In relation to the comment related to the pheromone dose-response for WT and *sst2*-deleted strains, we have now included a sentence on the rationale for moving forward with *sst2*-deleted strains, as well as indicated SST2 function.

"The rationale for deleting SST2 was to remove negative feedback regulation and increase sensitivity to the synthetic pheromones, as reported for native pheromones (Chan and Otte 1982)(Suppl. Fig. 1B)"

and

“The strains further contained deletion of the negative G protein signaling regulator SST2 which catalyzes the conversion of GTP to GDP in the G_α subunit preventing further signaling, as well as....”

p. 6 “expression should ideally be high in the absence of ligand and subsequently decrease upon ligand recognition to facilitate Gbeta/gamma release” – this statement makes no sense, Gbeta/gamma is released upon receptor activation regardless.

>> We agree that the formulation is unclear and have changed the small section from:

“Generally, while most hGPCR studies in yeast use constitutively active promoters to drive G_α expression in biosensing yeast^{6,8,25}, G_α expression should ideally be high in the absence of ligand, and subsequently decrease upon ligand recognition to facilitate G_{βγ} release and enhanced signal transduction. We identified two such dynamically regulated promoter candidates, P_{YIL169C} and P_{HFF1}, among the 57 core DEGs for strains expressing A2bR and MT₁ (Fig. 1D, Suppl. Table S2).”

to:

“..and we identified several DEGs relevant for future engineering of biosensing and actuation, such as AGA2, KAR4, and MFA1/MFA2, which all showed >10-fold activation across all strains.”

p. 7 “snipper” not “sniper”

>> No, this should be “sniper”. It is intended to refer to the precision of the approach. We decided to leave as is.

In addition to these responses, we also wish to highlight that with our new data in the revised manuscript we have also updated Methods descriptions and inserted plasmids, strains, and oligos in Suppl. Tables S5-S7. Note: Replaced CPK448 for CPK508. Text updates include:

“Co-transformation of these three fragments and the gRNA plasmid pDAM3 into strains CPK1, CPK4, and SBY4 replaced the native FUS1 open reading frame with yEGFP to give CPK86, CPK16, and SBY16, respectively. CPK86 was genetically deleted for SST2 as described for CPK88 to make strain CPK503.”

“pDAM123 and pDAM236-238 were made by assembling fragments from gDAM15 (α-leader secretion signal and P-factor) amplified with DAM76+DAM78 and P_{TEF1} (1564+1565), P_{FUS1} (DAM596+DAM655), P_{AGA2} (DAM598+DAM656), or P_{MFA1} (DAM657+DAM658) into pCfB2899.”

“P-factor producing strains were created by integration of NotI-digested pDAM123 or pDAM236-238 using gRNA plasmid pCfB3020 into CPK152 to make CPK508-511. NotI-digested pDAM123 was integrated in the same way into SBY55 to make SBY155.”

Reviewer #2 (Remarks to the Author):

Jensen et al presented the rational design of heterogeneous GPCR-based cell-cell communication system and used it to control mating pathway activation in *S. boulardii*. They first explored the transcriptomic changes when replacing the wild-type GCPR with four heterogeneous ones, and then engineered the *S. boulardii* based on the knowledge of *S. cerevisiae*, found the morphologies were regulated by four different pheromones, eventually constructed synthetic hGPCR-based cell-cell communication system and to control yeast mating pathway. It was an interesting topic to explored the new intercellular signaling pathway in a probiotic yeast (*S. boulardii*), even though the same topic had been extensively studied in both bacterium and yeast (Nature communications, 2018, 9(1): 1-12; Nature communications, 2020, 11(1): 1-11). Before considering to be published, I have several major concerns about the manuscript.

>> We thank the reviewer for reading and reviewing our manuscript. Please find our responses to all raised points below.

Regarding the prior art in *Nature Comm.*, we find limited similarities with our current study. The study by Billerbeck *et al.* presents an excellent toolbox to control yeast quorum sensing / synthetic cell-cell communication via fungal GPCRs and cognate peptide ligands. The distinctions to our current study is that Billerbeck *et al.* do not investigate human GPCRs, synthetic yeast mating, or GPCR-signaling in probiotic yeast. Du *et al.* also develops quorum sensing systems, for which they apply directed evolution of transcription factors. Du *et al.* differs greatly from our study by not investigating yeast mating or MAPK-signaling, and by employing intracellular transcription factors rather than GPCRs sensing the extracellular environment.

1, In line 53-57, “investigations of the compatibility of hGPCR with yeast mating have been limited....” need to give the proper reasons of the transcriptomic investigation for artificial system. The following design of the synthetic mating pathways seems not use the transcriptomic data.

>> This is a valid point. And we agree that we did not provide a clear connection between RNAseq and our follow-up engineering of synthetic mating, as otherwise indicated with our motivation statement:

“To elucidate transcriptome perturbations in yeast during hGPCR-signaling, and further guide efforts to successfully coupling of hGPCRs to the yeast mating pathway, we compared transcriptomes,..”

With our new synthetic mating data presented in Fig. 5G, we now take full advantage of the transcriptomics data by picking candidate DEGs from Fig. 1C-D as choices for promoters to drive inducible synthetic pheromone productions, akin to the regulation in native pheromone regulation. Importantly, this improves synthetic mating efficiencies. Please find our updated Fig. 5 in the revised manuscript, also documenting significant improvement of mating efficiencies. We thank the reviewer for pointing out the missing connection between RNAseq and devised synthetic mating trials, and we hope the updated new data clarifies the reasoning for the RNAseq study as well as its applicability for designing optimised synthetic matings.

2, In line165-167, why just estimated A2bR, MT1, and Mam2, but not involving 5-HT4b. the author need to present their reasons, since the following Figure still used the P-factor and 5HT4b system.

>> Valid point. Reviewer 1 also raised this point. For the main reason to study mating pathway activation using hGPCRs in diploid *S. boulardii*, we hope the reviewer agrees with us that the demonstration documented with human MT1 and A2bR sufficiently justifies our rational and conclusions. In the rational for our study we have now included the following text, which we hope addresses the reviewer's uncertainty related to our wish to showcase that mating pathway activation in two yeasts rather than one:

"However, in diploid S. boulardii, hGPCR-signalling in S. boulardii has never been reported, even though this could have major implications to advance the engineering of yeast therapeutics (13), especially as novel theranostics enabling both GPCR-based diagnosis as well as in situ therapeutics delivery based on an activate mating pathway."

The reason why we move on with all 4 hGPCRs for Figures 3-5, is to showcase more versatility in terms of the design space.

3, In line 195, the data of FSC-A was not shown in Figure 3A.

>> We thank the reviewer for paying attention to the distinction between FSC-A and SSC-A. We agree that the Fig. 3 reference was not clearly stated for SSC-A and changed the sentence from

"...and SSC-A ($R^2 = 0.92$) than with forward-scatter (FSC-A) ($R^2 = 0.81$) (Fig. 3, see Methods),"

to

"...and SSC-A ($R^2 = 0.92$) (Fig. 3A) than with forward-scatter (FSC-A) ($R^2 = 0.81$) (see Methods)".

4, In line 296-297, Why chose Serotonin and P-factor to construct the autonomous yeast mating system in Figure 5E, but only shown the serotonin, Melatonin and Adenosine three synthetic pheromones in Figure 5A-5D, not mention the data about the P-factor. The logic of the given data is weird, and uneasy to be following.

>> In Fig. 5 we achieve two milestones: i) full synthetic mating and ii) autonomous mating. To demonstrate full synthetic mating we tested the entire factorial design space for all human GPCRs in our repository, which agrees well with our transcriptome analysis strategy used in Fig. 1. Nevertheless, semi-synthetic mating efficiencies presented in Fig. 4 shows that P-factor and serotonin sensing strains perform best, so we decided to use these strains in our most endearing yet risky trial - that is autonomous mating. Our initial mating efficiencies for autonomous mating appeared suboptimal despite our choice of ligand-receptor pairs, which we now make up for by a) ratio tuning of the two mating strains and b) dynamic regulation of P-factor in these trials.

iii) Further, because P-factor is the only peptide used in our synthetic mating trials, we believe that the quick response required of dynamic regulation during a 5 hrs mating trial was more optimal with a peptide, such as P-factor, as compared to dynamic regulation of serotonin or melatonin.

We hope this response clarifies the logic behind the conducted experiments.

5, The colors and the sizes of triangle, square and circle is not easily distinguished with each other.

>> Good point. We have now increased the size of labels by 50% and increased contrast of the colours (see e.g. Fig 3E).

Reviewers' Comments:

Reviewer #1:

Remarks to the Author:

The authors have attempted to respond to my previous concerns. However there are major issues that are unresolved.

1. I don't think the approach is innovative. Heterologous expression of GPCRs in yeast was first done 30 years ago, and many times since.
2. I don't think the findings are biologically significant. I still don't understand how the work contributes to the development of probiotics or to our understanding of GPCR biology.
3. I don't think the data are reliable. GPCRs are pharmacological targets, and dose-response curves are standard in GPCR papers. Many of the dose-response curves presented in this paper are not publishable and look nothing like the positive control data for the native ligand-receptor system shown in Fig. S1B. In Fig S1C there is a very small (2-fold) induction at intermediate concentrations but no induction at high concentrations (!). In Fig. S2 the authors infer that the EC50s are 10-100x higher for receptors expressed in yeast than in mammalian cells. This disconnect in function is a concern. In addition, I don't see how anyone can obtain accurate EC50 values from so few data points, obtained at 10 fold increments of ligand concentration, and with such a poor signal. The heat map (Fig. S1D) looks promising but here I don't see how it is possible to obtain such clean data from such weak signals.
4. I remain concerned that the ligands could have an effect on transcription that has nothing to do with the expressed human GPCRs. The authors still provide incomplete transcriptomics data, do not provide a larger field of view for the cells shown in Fig. 3D (this could be placed in the supplement!) and did not do MAPK activity assays as a way to corroborate their findings.

Reviewer #2:

Remarks to the Author:

The authors have responded all my concerns.

Rebuttal to reviewer's comments:

1. I don't think the approach is innovative. Heterologous expression of GPCRs in yeast was first done 30 years ago, and many times since.

>> hGPCR expression in yeast has indeed been demonstrated in yeast at several occasions over the past decades. We claim novelty on i) first demonstration on the use of human GPCRs for yeast mating, for which we further demonstrate control by ligand supplementation (Fig. 4&5), and ii) that we can control autonomous mating with heterologous GPCRs by engineering dynamic expression of a synthetic pheromone (Fig. 5G).

2. I don't think the findings are biologically significant. I still don't understand how the work contributes to the development of probiotics or to our understanding of GPCR biology.

>> Developing biosensing yeast probiotics is an active field of research which attracts broad attention (Scott et al., Nat. Med., 2021, <https://pubmed.ncbi.nlm.nih.gov/34183837/>), and Fig. 2 asterisks indicate statistical significance of our results. Because *S. boulardii* is generally considered to have probiotic properties, whereas *S. cerevisiae* is not (Pais et al., J Fungi (Basel), 2020, <https://pubmed.ncbi.nlm.nih.gov/32512834/>) endowing probiotic yeast with biosensing capability is both novel and relevant.

3. I don't think the data are reliable.

- a) GPCRs are pharmacological targets, and dose-response curves are standard in GPCR papers. Many of the dose-response curves presented in this paper are not publishable and look nothing like the positive control data for the native ligand-receptor system shown in Fig. S1B. (...) In Fig. S2 the authors infer that the EC50s are 10-100x higher for receptors expressed in yeast than in mammalian cells. This disconnect in function is a concern. In addition, I don't see how anyone can obtain accurate EC50 values from so few data points, obtained at 10 fold increments of ligand concentration, and with such a poor signal.

>> We agree that dose-response curves are indeed standard in GPCR research, as well as using dilution series ranging at least 5 orders of magnitude as we and others do (Shaw et al., Cell, 2019 <https://pubmed.ncbi.nlm.nih.gov/30955892/> figure 4D for the same receptor, Mam2, as used in our study; Billerbeck et al., Nat. Comms, 2018 <https://pubmed.ncbi.nlm.nih.gov/30498215/> figure 2B). Still, the mere purpose of our dose-response curves presented in Fig. 1B is to recapitulate GPCR functionality and look at trends in our genetic backgrounds, before venturing into the story's innovative claims related to synthetic mating. Initially, we did not put in EC50s, but only following request from the Reviewer #1.

Furthermore, for expression of engineered hGPCR signaling in yeast cells there is no one-to-one design principle which will ensure dose-response curves for hGPCR will look alike the dose-response curves of native systems – whether in mammalian or fungal cells (Scott et al., Genetics, 2019 <https://pubmed.ncbi.nlm.nih.gov/30514708/>; Kapolka et al., PNAS, 2020 <https://pubmed.ncbi.nlm.nih.gov/32434907/>; Lengger et al., ACS Sensors, 2022 <https://pubmed.ncbi.nlm.nih.gov/35452231/>). It can be approximated through engineering, but depends on multiple factors, like G protein expression balancing, biased signaling, and reporter promoter used. However, it has not been an attempt of ours to exactly replicate human or native GPCR dynamics as translational models – instead we seek to functionalize the human receptors in a yeast context in regards to controlling and modulating yeast behavior, as well as optimizing GPCR performance as synthetic components. Likewise, the reported EC50 values of P_{FUS1}-activation are not of particular interest for the scope of the method development, as the main point that we investigate is whether the GPCRs are functional in activating the mating pathway, which relies on the regulation of a multitude of genes as seen from the transcriptome data from Fig. 1.

Furthermore, the GPCRs are investigated over a range of concentrations throughout the study, and we do not seek or rely on a single “optimal” concentration based on EC50 to achieve improvements seen for example in Fig. 4 & 5. To Fig. 1B we have now added a two-way ANOVA with a Dunnett's test (analysis was inserted in Suppl. Table S10) to directly assess increase in responses compared to the 0 μ M state for each strain design combination of G α -expression

level and GPCR, as it more clearly conveys the main point of the hGPCRs significantly activating the mating pathway, rather than relying solely on regression curves.

Regarding the appearance of our dose-response curves we have presented them in engineered systems spanning 5 orders of magnitudes in ligand conc. (Fig. 1B). For designs with tight stoichiometric control of G protein release (i.e. pPGK1 promoter:Gα) we have sigmoidal curves for ¾ dose-response curves. For Mam2 the curve does not display sigmoidal behavior for the concentrations applied, presumably because our system is highly sensitive and reaches MAPK pathway overstimulation and hence a decreased signal at 100 μM of P-factor - which is anyway not a physiologically relevant concentration. Others, such as Shaw et al., Cell, 2019 cited above, do not include the 100 μM measurement for Mam2, which we further assume is due to its detrimental effect on the commonly applied variable slope (four parameter) nonlinear regression (Shaw et al., Cell, 2019 <https://pubmed.ncbi.nlm.nih.gov/30955892/>). Accordingly, we do indeed see lower mating efficiencies for very high concentrations of ligands (Fig. 4C & 4E, Fig. 5C-F). Yet both for the sake of consistency and to include this interesting biological readout, we included this highly reproducible observation. Regulatory mechanisms for pathway overstimulation and negative feedback have previously been reported in relation to MAPK component phosphorylation, which could explain the observed dynamics (Anders et al. 2020, <https://pubmed.ncbi.nlm.nih.gov/32661402/>).

As for the reviewer's concern on the native ligand-receptor pair shown in Fig. S1B, we observe the same general pattern as in Fig. 1B - that increasing dosages of ligand increases the reporter output, except that overstimulation is never reached for the native system in the chosen range of supplemented ligand.

We hope that this reasoning clarifies the chosen range of supplemented ligand concentrations and disconnect between mammalian and yeast GPCR studies.

b) In Fig S1C there is a very small (2-fold) induction at intermediate concentrations but no induction at high concentrations (!).

>> This is indeed a peculiar trait specific to this 5-HT_{4b} strain balanced with the RNR2 promoter driving expression of Gα. We believe there is still much to be understood about this receptor.

For example, introducing 5-HT_{4b} into the RNR2p-G α strain results in >10-fold decreased background signaling of the GFP reporter, which is only seen with this receptor (Suppl. Fig. S1E). However, it is exciting that we observed a 2-fold increase in this admittedly narrow range between 1 μ M and 10 μ M of supplemented ligand. We hope that the reviewer agrees that the small inductions observed in Suppl. Fig. S1C are due to the peculiarity of 5-HT_{4b} and not a result of poor execution or unreliable data. Indeed, we observed high reproducibility between replicate experiments (Suppl. Fig. S1C).

c) The heat map (Fig. S1D) looks promising but here I don't see how it is possible to obtain such clean data from such weak signals.

>> The signals that we report for the heatmap in Suppl. Fig. S1D reaches 90-fold induction with ligand over background for cognate ligand-receptor pairs, while cross-reaction between non-cognate ligand-receptor pairs is nearly non-existing. We have now included the raw data for the matrix and apologize for not including this Source data in our first resubmission. Please see Supplementary Fig. S1D in the revised Source Data file. We also inserted the data table further below in this rebuttal for convenience. We hope the reviewer will acknowledge that not only are our obtained signals not weak, but also that the quality is high as inferred from the reproducibility of signal strength obtained between the replicates. Furthermore, the obtained results for melatonin receptor corroborate previous findings, namely MT₁ is specific for its cognate ligand and only minimally responds to the tryptanergic precursor of melatonin, serotonin (Shaw et al., Cell, 2019 <https://pubmed.ncbi.nlm.nih.gov/30955892/>, figure 6A). We hope the raw data and benchmark to existing prior art address the concern raised by the reviewer.

4. I remain concerned that the ligands could have an effect on transcription that has nothing to do with the expressed human GPCRs. The authors still provide incomplete transcriptomics data, do not provide a larger field of view for the cells shown in Fig. 3D (this could be placed in the supplement!) and did not do MAPK activity assays as a way to corroborate their findings.

a) I remain concerned that the ligands could have an effect on transcription that has nothing to do with the expressed human GPCRs.

>> Ok. Maybe we were not clear enough in our previous reply to this point. We are basically being questioned whether the ligands used in this study themselves would impact the presented transcriptomes. Our experimental 60-sampled design did not address this question. However, because we have a multi-factorial design (ie. dosage x genotype) we were able to enumerate how many genes are differentially expressed in hGPCR-expressing cells compared to *STE2*-expressing cells in the absence or presence (any conc.) of ligands. This data we already provided, and now we also submit the Source data file for this analysis as part of our resubmission (see Source Data Fig. 1C&D or further below in rebuttal). This file shows that none of the reporter promoters used in this study are DEGs when solely looking at transcriptomes +/- ligands for any of the three hGPCR strains. Further to this, we wish to highlight that included in our “cross-reaction or specificity-matrix” in the first revision of the manuscript (Suppl. Fig. S1D + new Source Data file Suppl. Fig. S1D) we include the requested negative control data – both “no ligand” and “no GPCR” controls, showing that *FUS1* promoter is not regulated by any of the used ligands. As this promoter is used, now among others presented in our revised manuscript, to drive synthetic mating, we hope the reviewer will agree that further justification for choice of promoter(s) to be suitable for full synthetic mating is out of the scope of the presented study.

b) The authors (...) do not provide a larger field of view for the cells shown in Fig. 3D (this could be placed in the supplement!)

>> We now provide a larger field of view in our revised Supplementary Figure S3D. Here we present many more cells of representative morphologies of engineered cells expressing either of the 4 hGPCRs co-expressed with *Gα* controlled by the strong *PGK1* promoter. As is evident from these micrographs there is no shmooing in cells without the hGPCR ligand being present. We hope this new addition addresses the comment from the reviewer on too few cells being shown in the original manuscript.

c) (The authors...) did not do MAPK activity assays as a way to corroborate their findings.

>> This is true. Previous studies have indeed used the same P_{FUS1} -GFP as a reporter for MAPK activation and importantly specifically for assaying the activity of the involved phosphatases and kinases (Anders et al. 2020, <https://pubmed.ncbi.nlm.nih.gov/32661402/>). Also, we would like to highlight the already referenced studies of our first rebuttal. The yeast pheromone pathway is an archetypical MAP kinase signaling route, with lots of prior art connecting the Fus3 MAP kinase to phosphorylation of the transcription factor Ste12 directly controlling expression of Fus1 and many other genes. We have searched for studies showing the activation of Fus3 kinase and the other kinases upon activation of hGPCRs in yeast, but have not found the example. This said, we ultimately rely on Ste12 activation, only to happen upon phosphorylation by Fus3 or Kss1, and as our study is a method, not a descriptive study of pathway signaling, we would hope the reviewer would focus on the successful demo of a new method rather than a focus on whether a MAP kinase assay is presented as an interim description of what is generally perceived as well-established in the yeast-GPCR field.

Cross-reaction matrix

Source Data Suppl. Fig.

S1D:

	Ste2	MT1	Mam2	A2bR	5-HT4b	No GPCR
	794723,5	5584	7412	8252	16863	5731
α-factor	778410	5540	7214,5	8323	16905	5716
	765386	5506	7209	8205,5	16907	5818
	15640	69177,5	6729	7663	16366	5503
Melatonin	15743	71378,5	6746	7671	16822	5514
	15501	68327	6787	7631	16421	5560

	17240	5752	628977,5	8172	17216	6070
P-factor	17543,5	5771	590471	8398	17350	6042,5
	17716	5821	650395,5	8338,5	17527	6158
	20124	6081,5	6755	763678,5	14847	6523
Adenosine	19546	6116	7058	706248	14861	6674
	20194	6068,5	6958,5	574618	15137	6665
	15905,5	5265	6814	7762,5	330154	5695
Serotonin	16214	5266	6850	7686	289747,5	5789,5
	15923	5408	6943	7766	310022	5926
	20092	6007	6916	7916	14941	6683
No ligand	20097,5	6061	7005	7872	15941	6540
	19895	6243	7068	7817	15419,5	6758

Source Data Fig. 1C&D:

Gene name	MT1_wo_ligands	MT1_ligands	5HT4b_wo_ligands	5HT4b_ligands	A2bR_wo_ligands	A2bR_ligands
YBL053W		UP				
YPL185W			UP			
YIL012W						
HHY1						
MT1 NOG2			UP	UP	UP	UP
10 genes YGR045C						UP
YNL150W						
YCL048W-A						
YER189W						
ATC1			UP			
YMR173W-A			DOWN	DOWN		DOWN
YOR289W			DOWN	DOWN		
YBR241C			DOWN	DOWN		DOWN
SUE1			DOWN	DOWN	DOWN	DOWN

5HT 4 53 genes	YIL024C			DOWN	DOWN		
	MET28	UP		DOWN	DOWN		DOWN
	LEE1			DOWN	DOWN		
	ATG33			DOWN	DOWN		DOWN
	YJL163C			DOWN	DOWN	DOWN	DOWN
	SDS24			DOWN	DOWN	DOWN	DOWN
	THI5			DOWN			
	PAU7			DOWN			
	SAM3			DOWN			
	AI2			UP	UP		
	YMR00						
	1C-A	UP	UP	UP	UP		
	SRT1		UP	UP	UP		
	YOR248						
	W	UP	UP	UP	UP		
	AI1			UP	UP		
	LSO1			UP	UP		
	BSC5			UP	UP		
	YLR285						
	C-A	UP	UP	UP	UP		
	TMA17				DOWN		DOWN
	PET10				DOWN		
	CYC3				DOWN		
	CAT8				DOWN		
	YOR186						
	W				DOWN		
	YNR014						
	W				DOWN	DOWN	DOWN

YDR535 C		DOWN	
YKL133 C		DOWN	DOWN
GDH1		DOWN	
OAR1		UP	
YJR146 W		UP	
YOL164 W-A		UP	
FRM2		UP	UP
YIG1		UP	UP
YCR025 C	UP	UP	
YDR183 C-A		UP	
YLR122 C		UP	UP
PDC6			
PAU12			UP
AMN1			
NKP2			
GPB1			
YDR344 C	UP		UP
YRF1-2			
ANB1			DOWN
REC104			
YTP1			
DCI1			DOWN

Statistics		Specific DEGs	DEGs in 0uM	DEGs in ligands
MT1	10	0	1	
5HT				
4	53	21	35	
A2b				
R	13	1	8	

Reviewers' Comments:

Reviewer #1:

Remarks to the Author:

The authors have done an admirable job responding to my concerns. My reaction to the key points:

1. OK. This is a matter of opinion, what I find innovative does need to match that of the authors or the readers.
2. OK. This is a matter of opinion, what I find significant does need to match that of the authors or the readers.
3. Not OK. I still think the authors should provide more complete dose-response curves, as done in the papers cited by the authors (Kapolka, Billerbeck, Shaw, etc.) These are very simple experiments, would take a week or two at most, are standard in the field, and would increase confidence in the findings. I think it is dangerous to rely on a very small increase in only 2 data points as done in Fig. S1C, S2A, S2C, and doing so would provide more accurate EC50 values as done in the papers cited.
- 4a. OK. This is a matter of opinion, what I find to be an important control does need to match that of the authors or the readers.
- 4b. Not OK. The authors still do not provide a larger field of view for the "representative" cells shown in Fig. 3D, these are different cells.
- 4c. OK, I agree too few labs do this orthogonal assay, that would corroborate the findings (point 3 above), but they should. It is more proximal to the GPCR and would corroborate the transcription measures reported here.

Rebuttal to reviewer's comments:

1. I still think the authors should provide more complete dose-response curves, as done in the papers cited by the authors (Kapolka, Billerbeck, Shaw, etc.) These are very simple experiments, would take a week or two at most, are standard in the field, and would increase confidence in the findings. I think it is dangerous to rely on a very small increase in only 2 data points as done in Fig. S1C, S2A, S2C, and doing so would provide more accurate EC50 values as done in the papers cited.

>> We appreciate this opportunity to increase confidence in our dose-response assays. We now provide curves which span up to 13 different concentrations for all the *S. cerevisiae* strains (new Suppl. Fig. S1D) that were previously used for downstream mating trials and transcriptome analysis (P_{PGK1} - G_{α}), and which relate to Fig. 1B. These curves were used to calculate new EC50 values (updated Suppl. Table S1). Furthermore, and surprising to us, the two data points of serotonin concentrations previously observed to show reporter gene expression were not reproduced following several rounds of replicate experiments, and the original Fig. S1C has therefore been deleted as it does not provide new information compared to the original main Fig. 1B. The main text has been updated accordingly:

“In accordance with previous studies ^{8,11,21}, we observed that the P_{FUS1} -GFP reporter was induced in all biosensing strains across a gradient of cognate ligand dosages (Fig. 1B), except no signaling was observed for 5-HT_{4b} in the P_{RNR2} -GPA1/ $G_{\alpha 12}$ strain design (Fig. 1B, ~~Suppl. Fig. S1C~~). Importantly, and corroborating previous work on fungal GPCRs ⁷, hGPCRs are orthogonal across their non-cognate ligands when expressed in yeast (Suppl. Fig. S1C), while for strain designs with the strong promoter PGK1 controlling expression of the G_{α} protein the EC50s observed for MT₁ and 5-HT_{4b} are approximately two orders of magnitude higher in yeast cells, and approximately 2-fold lower for A2bR in yeast cells, compared to mammalian cells (CHO)(Suppl. Table S1; Suppl. Fig. S1D).”

Finally, we updated Suppl. Fig. S2A and S2B with an additional 1000 μ M ligand concentrations for adenosine and melatonin, thus spanning six orders of difference in ligand concentrations.

For Suppl. Fig. S2C this last concentration was left as this concentration exceeds the solubility of P-factor. We believe that adding more increments here would not change the standing conclusions, namely that even though the *sst2* deletion only seem to increase sensitivity of the mating pathway in *S. cerevisiae*, main Fig. 2 shows the first observation that in engineered *S. boulardii* the mating pathway can be activated by hGPCR.

Taken together, we hope the reviewer agrees that for the strains used for characterizing and engineering synthetic mating in *S. cerevisiae*, the 13-data points dose-response curves are now more complete, just as extra data points for *S. boulardii* further illustrates *S. boulardii* mating pathway saturation.

2. The authors still do not provide a larger field of view for the "representative" cells shown in Fig. 3D, these are different cells.

>> We are sorry for the misunderstanding here, and thank the reviewer's patience. The original pictures were taken for highly dilute cultures, and including the whole frame did not sufficiently address the reviewer's request. We now updated Suppl. Fig. S3D with pictures from cultures with increased densities and include all strains shown in main Fig. 3D to provide a thorough representation of phenotypes, now highlighted with arrows as well.